

# Yeti 1.0: a generalized framework for constructing bottom-up emission inventory from traffic sources

Edward C. Chan[1], Joana Leitão[1], Andreas Kerschbaumer[2], Timothy M. Butler[1]

[1]Institute for Advanced Sustainability Studies, Potsdam, Germany
[2]Senatsverwaltung für Umwelt, Mobilität, Verbraucher- und Klimaschutz, Berlin, Germany

*Correspondence to*: E.C. Chan (edward.chan@iass-potsdam.de)

**Abstract.** This paper outlines the development and operation of Yeti, a bottom-up traffic emission inventory framework written in the Python 3 scripting language. A generalized representation of traffic activity and emission data affords a high degree of scalability and flexibility in the use and execution of Yeti, while accommodating a wide range of details on
topological, traffic, and meteorological data. The resulting traffic emission data are calculated at a road level resolution on an hourly basis. Yeti is initially applied to traffic activity and fleet composition data provided by the Senate Administration for the City of Berlin, which serves as the region of interest, where the Yeti calculated emissions are highly consistent with officially reported annual aggregate levels, broken down according to different exhaust and non-exhaust emission modes. Diurnal emission profiles on select road segments show not only the dependence from traffic activities, but also from road
type and meteorology. These road level emissions are further classified on the basis of vehicle categories and Euro emission classes, and the results obtained confirmed the observations of the City of Berlin and subsequent rectifications.

## 1 Introduction

Accurate quantification of emission sources is a primary determinant to establishing the relevance and trustworthiness of air quality model results (Thunis et al, 2016). However, this presents a key challenge in the construction of emission inventories
from vehicle traffic, owing to the number and distribution of individual pollutant sources, as well as a plethora of factors influencing their output (Davison et al, 2021). These factors can be technological – such as powertrain, and emission control – meteorological, topological, as well as behavioral – including but not limited to ambient temperature, road and traffic conditions. That vehicles release pollutants in motion and at rest at irregular intervals introduces further complexity, and therefore the culmination of these considerations must be conducted at high temporal and spatial resolutions, as the application
of traffic emission inventory based on static, annual mean activity data could lead to large discrepancies between model results and observation data in pollutant concentration (Kuik et al, 2018), due to non-linear relationships between traffic flow and emission levels (Tsanakas, 2019).



Traffic emissions inventories can be obtained with either spatial-temporal redistribution of aggregated traffic activity data values – the so-called "top-down" approach – or by integration of vehicle-level emission factors estimated from existing databases, otherwise known as the "bottom-up" approach, to the street-level resolutions. While the top-down approach requires knowledge of existing aggregated emissions data – such as total fuel consumption – and adapting them to regional fleet composition – the bottom-up approach relies on vehicle-level emissions data and is less sensitive to atmospheric transport processes and boundary conditions (Gurney et al, 2017), thus providing a more representative data set for modelling air quality for the region of interest. Moreover, the bottom-up methodology also affords emissions for different scenarios to be conceived, designed and explored, independent of historical aggregate levels, allowing exploration of alternative scenarios due to, for instance, adaptation of new policies, or shifts in public perception and preferences towards urban mobility (Kollosche et al, 2010) as well as changes in public infrastructure (Schmitz et al, 2021). These existing or future traffic scenarios can be compiled, for example, through surveillance (Buch et al, 2011), stochastic parameterization (Thonhofer and Jakubek, 2018), or agent-based modelling (Seum et al, 2020). The heterogeneity of incoming data must therefore be considered in designing and developing a traffic emission inventory methodology.

Meanwhile, traffic emission data are generated using a composite of sources. Time resolved traffic activity data over the road network, namely traffic flow and fleet composition, are derived either from direct observations conducted at key locations of the region of interest – for example using license plate recognition (Schmidt and Düring, 2016; 2021) – or inferred from surrogate data such as representative diurnal cycle (Builtjes et al, 2003) and peak traffic flow data (Ibarra-Espinosa et al, 2018). This gives rise to the counts of different vehicle classes driving through each link in the road network at any given time, as well as the corresponding traffic condition subject to the function of the road link and capacity. In turn, the emission output of each vehicle travelling through each road link can be calculated using emission factors specific for different operating and traffic situations, which can then be aggregated over the region of interest on an annual basis. Frameworks for emission factors are available, with the Computer Programme to calculate Emissions from Road Transport (COPERT; Ntziachristos et al, 2009), the Motor Vehicle Emission Simulator (MOVES; US-EPA, 2021), and the Handbook for Emission Factors for Road Transport (HBEFA; INFRAS, 1999). Meteorological and seasonal effects, that is, ambient temperature and fuel blend vapor pressure, also have a significant impact on cold-start and evaporative emissions. They can be incorporated into the emission inventory calculation.

Despite the details and resolution that the bottom-up approach can provide, and the extent of information required to generate the emission data, traffic emission inventories, particularly those presented at an official capacity in Germany, are only reported in annual aggregate levels, typically to be compliant with existing guidelines, such as those set forth by the Society of German Engineers (Verein Deutscher Ingenieure; VDI, 2020), exemplified by Diegmann et al (2020) and Herenz et al (2020). Other bottom-up emission inventory tools are available, such as the COPERT-based Vehicular Emissions Inventory library package (VEIN; Ibarra-Espinosa et al, 2018) – as deployed in the coupled Eulerian-Lagrangian dispersion study of Veratti et al (2020) – the Spatial Regression Truck model (SPARE-Truck; Perugu et al, 2017) for heavy-duty vehicles, each operating with a



specific set of traffic or surrogate activity data, as with the High-Elective Resolution Modelling Emission System (HERMESv3; Guevara et al, 2020). Proprietary software frameworks, such as the HBEFA-based IMMIS/em (Diegmann, 2008), are also available, but the cost and the level configurability play a significant role in their application for open scientific exploration of traffic scenarios, as a standalone tool, or as part of an evaluation and modelling toolchain.

This study introduces a framework – Yeti – a HBEFA-based traffic emissions inventory written in the Python 3 scripting language, which adopts a generalized treatment for activity data, such that inventories can be created using traffic information of varying levels of detail introduced in a systematic and consistent manner. More importantly, as preprocessed input data such as road network traffic conditions and emission factors are large and require copious time and computational effort to compile and generate, the ability to maximize reusability is also a critical design consideration, so that emission data can be generated

under different configurations on already available preprocessed input data. As a result, Yeti has been conceived and implemented with a high degree of data and process symmetry, allowing scalable and flexible execution while affording ease of use.

In collaboration with the Senate Administration for Environment, Mobility, Consumer and Climate Protection (Senatesverwaltung für Umwelt, Mobilität, Verbraucher- und Klimaschutz) for the City of Berlin, the emission data generated

by Yeti are evaluated using official aggregate inventory values, where particular emphasis is placed on carbon monoxide (CO), unburnt hydrocarbons (HC), nitrogen oxides (NOx), and particulate matter (PM) for the current study. An examination of emissions at road level is followed to investigate the contributions of different vehicle sectors to local traffic emissions, in order the demonstrate the versatility of Yeti as a standalone tool for investigating emission source attribution, or as an integral part of an existing air quality modelling tool chain, such as the Weather Research and Forecasting model with Chemistry

(WRF-Chem; Grell et al, 2005) or OpenFOAM (Weller et al, 1998; Chan and Butler, 2021).

## 2 Model description

The basic premise of Yeti is to produce hourly pollutant emissions (e.g., NOx, CO, and PM) from traffic sources under different ambient conditions, over a collection of road segments with information on traffic count and fleet composition resolved at a road level. Geometrical attributes for the road segment – length, grade, and traffic directions – are used as topological input

data. Emission factors for each vehicle subsegment are read from user supplied HBEFA tables. Diurnal temperature profiles and Reid vapor pressure (RVP; $p_{RV}$) can also be provided to allow emission calculations to account for seasonal variations and local ambient meteorological conditions. The current version of Yeti derives emission values from the HBEFA emission factors according to vehicle subsegments.





## 2.1 Hourly emission calculation strategies

The hourly emissions for each pollutant species over each road link for all HBEFA vehicle subsegments can be summarized in Eq. (1) below:

$$E_n^l = \{\sum_{\forall k}[e_H^l + e_C^l + e_D^l + e_S^l + e_R^l + e_N^l]_k\}_n \tag{1}$$

where $E_n^l$ is the hourly emission for pollutant species $(n)$ over road link $(l)$ summed over all vehicle subsegments $(k)$, $(e^l)_n$ is the contribution from the individual emission modes for species $(n)$ on road link $(l)$ and vehicle subsegment $(k)$ differentiated

by individual subscripts: $H$ denotes hot exhaust emissions, $C$ for cold excess exhaust emissions, $D$ for evaporative diurnal emissions, $S$ for evaporative hot soak emissions, $R$ for evaporative running loss emissions, and $N$ for non-exhaust PM emissions. Generally speaking, the emissions for each contribution are determined by multiplying the HBEFA emission factor for the corresponding vehicle subsegment $(\varepsilon^k)_n$ with a quantity corresponding to the traffic conditions, such as the number of vehicles belonging to the vehicle subsegment travelling through the road link $(N_k^l)$. Detailed descriptions for the calculation of

each contribution will be provided in the following sections.

### 2.1.1 Hot exhaust emissions

Hot exhaust emissions originate from the vehicle's tailpipe after the operation of the powertrain and exhaust systems have reached thermal stability. They are directly proportional to the total kilometer driven by the vehicles belonging to subsegment $(k)$, and it is expressed as the product between emission factors, the number of vehicles $(\Lambda_k^l)$ corresponding each level of

service (LOS; $\lambda$) indicating the traffic saturation level of the road link in question, and the length of the road link $(x_l)$ as indicated in Eq. (2):

$$(e_H^l)_n = \sum_\lambda [\Lambda_k^l \cdot x_l \cdot (\varepsilon_H^k)_n], \tag{2}$$

where $N_k^l = \sum_\lambda \Lambda_k^l$, that is, $N_k^l$, the number of vehicles belonging to subsegment $(k)$ travelling through road link $(l)$, is the sum of corresponding vehicles across all LODs $(\lambda)$.

Yeti identifies the hot exhaust emission factor for each pollutant and vehicle subsegment at specific road grade and traffic situation at the road link, which comprises the area (urban or rural), road type (motorways, trunk roads, distributors, access roads, etc.) and posted speed limit, as well as the corresponding LOS.

### 2.1.2 Cold excess exhaust emissions

Additional exhaust emissions can be accrued by the vehicle while it is transitioning from its initial "cold" state to the thermally

stabilized state described in Section 2.1.1. Cold excess exhaust emissions refer to the difference in exhaust emissions between the elevated emission level and the base hot exhaust emission level. The cold excess exhaust emission factors in HBEFA are defined as being dependent on the ambient temperature, warm-up time, and warm-up distance. However, the public version of



HBEFA only allows cold excess exhaust emission factors to be varied by one independent variable at a time, while keeping the other two at predetermined average levels. Thus, the current implementation of Yeti assumes cold excess exhaust emission factors as a sole function of temperature.

Further, in HBEFA, the cold excess exhaust emissions are scaled by the number of cold starts that take place on the corresponding road link. Data on cold start counts were not available in the traffic data provided to the authors. In light of this, the cold start count can be inferred by the hourly traffic count ($N_k^l$) as well as the road type. In Yeti this is represented by a dimensionless factor $\chi_C^l$, representing the fraction of traffic flow that are identified as cold start events, as shown in Eq. (3):

$$(e_C^l)_n = \chi_C^l \cdot N_k^l \cdot (\varepsilon_C^k)_n, \tag{3}$$

where $\varepsilon_C^k$ is the HBEFA cold excess exhaust emission factor for vehicle subsegment $k$. For the current study, the value of $\chi_C^l$ have been set to 0.3 for all collectors and access roads, which correspond to road types 40, 41 and 50 in HBEFA versions 3.3 and 4.1, and zero for all other road types.

### 2.1.3 Evaporative emissions

While combustion processes are the primary production mechanism for vehicular emissions, fuel and various volatile fluids can escape into the atmosphere through exposure. This type of emissions is known collectively as evaporative emissions, which can take place due to temperature fluctuation (diurnal, $e_D^l$), recent cessation of vehicle movement and engine operation (hot soak, $e_S^l$), or continuous leakage while the vehicle is in operation (running losses, $e_R^l$). HBEFA defines each of the evaporative emission factors under different diurnal temperature profiles and fuel RVP ($p_{RV}$).

Evaporative diurnal emissions are scaled by the number of vehicles for each vehicle subsegment ($k$). Since the emission factors ($\varepsilon_D^k$) are measured on a daily basis, the hourly redistribution of $\varepsilon_D^k$ can be estimated through the application the empirical relation of Landman (2001) on an hourly diurnal ambient temperature profile:

$$\widehat{\Phi}(h) = \beta_0 + \beta_1\left(\Delta T_{\min}^{h-1}\right) + \beta_2\left(\Delta T_{h-2}^{h-1} \cdot \Delta T_{\min}^{h-1}\right) + \beta_3 p_{RV}\left(\Delta T_{h-2}^{h-1}\right)^2 + \beta_4\left(\Delta T_{h-1}^{h}\right) + \beta_5 p_{RV}\left(\Delta T_{\min}^{h-1}\right), \tag{4}$$

where $\widehat{\Phi}$ is the day to hour redistribution factor, $h$ is the indicated hour in local time, $p_{RV}$ is the RVP, $\beta_0$ to $\beta_5$ represent the empirical single-valued constants tabulated in Table 1. Differences in hourly temperatures are expressed in shorthand notations where $\Delta T_{\min}^{h-1} \equiv T(h-1) - T_{\min}$, $\Delta T_{h-1}^{h} \equiv T(h) - T(h-1)$, and $\Delta T_{h-2}^{h-1} \equiv T(h-1) - T(h-2)$, for $T_{\min}$ is the minimum hourly ambient temperature of the diurnal cycle, $T(h)$, the ambient temperature at the hour of evaluation, and $T(h-i)$ indicates the ambient temperature at $i$ hours prior to the hour of evaluation. It should be noted that Eq. (4) applies to fuel-injected vehicles passing both purge and pressure tests, which through its usage Yeti implicitly assumes in the vehicle fleet composition. In addition, as Eq. (4) does not unconditionally exhibit properties of a probability weight function – that is, $\sum \widehat{\Phi}(h) = 1$ and $\widehat{\Phi}(h) \geq 0$ for all hours $h$ over the diurnal cycle – negative values of $\widehat{\Phi}$ are set to zero, after which the hourly $\widehat{\Phi}$ are normalized. The expression for the hourly evaporative diurnal emissions ($e_D^l$) then becomes:





$$(e_D^l)_n = N_k^l \cdot \widehat{\Phi}(h) \cdot (\varepsilon_D^k)_n. \tag{5}$$

On the other hand, hot soak emissions ($e_S^l$), are scaled with the number of engine stops in HBEFA, and the corresponding
emission factors are dependent on the seasonal changes in mean ambient temperature the Reid vapor pressure ($p_{RV}$). In a
similar treatment as cold excess exhaust emissions, the number of engine stop events are estimated – in the absence of direct
data – through the hourly traffic count ($N_k^l$) and road type. Thus the hourly evaporative hot soak emissions ($e_S^l$) can determined
with Eq. (6):

$$(e_S^l)_n = \chi_S^l \cdot N_k^l \cdot (\varepsilon_S^k)_n, \tag{6}$$

where, $\chi_S^l$ represents the fraction of traffic flow representing engine stops, in the same manner as for $\chi_C^l$ in Eq. (2).

In the meantime, in addition to ambient and seasonal conditions, evaporative running loss emissions ($e_R^l$) also depend on the
category of the road link, that is, whether it is motorway, rural or urban roads. In HBEFA, $e_R^l$ is scaled by the number of
kilometers driven, and is expressed in Eq. (7):

$$(e_R^l)_n = N_k^l \cdot x_l \cdot (\varepsilon_R^k)_n. \tag{7}$$

### 2.1.4 Non-exhaust particulate matter

In addition to evaporation, abrasion and resuspension of particles constitute the other form of non-exhaust emissions. More
precisely, abrasion refers to the process from which wear particles are generated through shear forces from the braking and
tire-road friction. On the other hand, resuspension refers to the reentrance of settled particles into the ambient air through wind
and passing vehicles and is not a source of new non-exhaust emissions (Vanherle et al, 2021). HBEFA provides a simplified
form for calculating non-exhaust particulate matter, in which estimates for emission factors are available without distinguishing
between contributions from abrasion and resuspension. As non-exhaust PM is generated when the vehicle is in motion, it
depends on the distance travelled by each vehicle, as shown in Eq. (8)

$$(e_N^l)_n = N_k^l \cdot x_l \cdot (\varepsilon_N^k)_n. \tag{8}$$

Further, as emission factors for non-exhaust PM are only of restricted availability in HBEFA 3.3, it has been made available
using emission factors from HBEFA 4.1 by remapping the vehicle subsegment ID to HBEFA 3.3 equivalents. The remapping
method is described in detail in the Supplemental document (Chan et al, 2022).

### 2.2 Implementation and structure

Yeti has been developed with the Python 3 scripting language. Standard modules are used to maximize portability and
compatibility. Multithreading is supported via the `concurrent.futures` module, and user-defined configurations are
specified using the `yaml` module. Input and output data files are processed tab-delimited tabulated text. The calculation of
different emission modes as described in Section 2.1, are referred to as strategies, which can be explicitly activated or





deactivated by the user. In addition, the traffic network of a given city often contains a large number of road links. Thus, an option is provided for Yeti to operate only on a subset of the traffic network to further reduce processing time and data size. This is useful when the emissions pertaining to the entire network is not required. The following sections describe the
180 organization of Yeti input and output files, required configurations, and program execution. Further details can be found in the supplementary document to this article.

### 2.2.1 Data organization

Figure 1 illustrates the organization of input and output data of Yeti. Each set of configuration, input, and output data are organized in separate directories known as run cases. The configuration of each run case is specified by the user through a file
in YAML format, which includes general system settings, locations of input and output datasets, as well as configurations for each emission mode. Input HBEFA tables, traffic flow data, and meteorological data are saved in different directories under each run case. Additional configuration options can be specified to ensure uniformity in HBEFA version in the emission data and the processed hourly traffic data.

Output data consists of summaries of run configuration, road link data, and vehicle subsegment definitions, as well as hourly
emissions for all specified pollutants for each HBEFA defined vehicle subsegment, sorted by road link (that is, the name of the road segment and traffic direction). The hourly emissions data are then tabulated for each vehicle subsegment and are categorized for each unique combination of pollutant type, emission strategy, day type and meteorological profile. Any existing output directory will be renamed to preserve already created data in the event of output directory name conflict.

### 2.2.2 User-specified configuration

The configuration file (config.yaml) for the run case defines the execution parameters and options used by Yeti. As mentioned previously, locations of various input files for HBEFA emission factors, traffic flow and meteorological data are summarized in the configuration. Further customizations are also possible to increase flexibility for accommodating a variety of nomenclatures and data naming conventions. Specific settings are also available to instruct Yeti to process only part of the road network by road link name or by street name to reduce runtime by restricting the problem size. Further, meteorological
data can be represented with either user-specified mean temperature and RVP, or when additional details are available, diurnal profiles of hourly ambient temperatures and corresponding seasonal RVPs.

In addition, each emission calculation strategy can be specified individually and independently. Once the strategy has been activated, HBEFA emission factor tables are to be provided by the user, as well as the pollutants which are to be calculated using the strategy. Provisions are also available for further customizations, such as indexing rules for emission factor tables,
or the fractions for cold starts ($\chi_C^l$) for cold excess exhaust emissions or engine stops ($\chi_S^l$) for evaporative hot soak emissions, although default values have been provided and need not be explicitly specified in the configuration file.



### 2.2.3 Execution flow

A flow chart for the general program flow for Yeti is presented in Fig. 2. Upon start of execution, Yeti locates the run case directory and begins reading and validating the run configuration, which determines correctness of user specification, as well
as the existence and integrity of all input data. Yeti then proceeds to locate and backup any existing emission output. Data logging will also be enabled at this point to write out notifications and diagnostic information. In the following steps Yeti continues to load all input data and proceeds to data compaction by removing entries in the HBEFA emission factor tables that are not used, for instance, for vehicle subsegments not present in the traffic fleet composition, or for pollutants that have not been specified in the user configuration. To further accelerate data look-up, each emission factor table is indexed by generating
hash keys based on unique combinations of data fields.

Once the input data have been properly prepared, the emissions are processed in a multithreaded environment. The hourly emission data for all vehicle subsegments are calculated at each road link for every active pollutant, activated emission strategy (for example, hot exhaust, cold excess exhaust, and non-exhaust PM) and day type. They are then written into tab delimited tabulated text files. A queue consisting of all road links are fed into a thread queue to manage the continuous workflow. Once
the emission calculations for all road links are completed, the thread pool is terminated, and the log file is finalized before the program ends.

### 3 Aggregate emission evaluation

As an illustrative example, a possible methodology for preparing Yeti input data is presented in the following sections, in conjunction with traffic source data made available at the discretion of the Berlin Senate. The Yeti emissions output are then
compared with the official annual aggregate figures for year 2015, produced in accordance with Guideline 3782, Leaflet 7 of VDI (2020), from the from the City of Berlin (Diegmann et al, 2020). The reader can refer to the supplementary document for further technical details and instructions for the execution of Yeti, as well as processing of input and output data.

### 3.1 Preparation of source input data

Yeti requires input data for hourly traffic flow, meteorological and seasonal data, as well as HBEFA emission factors. Each of
230 these data sets are accessible from independent locations, as illustrated in Fig. 1, where all end-point input and output data are stored in tab-delimited tabulated text files with header row labels. Yeti has been accordingly designed to provide some flexibility to accommodate diversity in source data format and content from which the Yeti input data described in Section 2.2.1 are derived. Detailed instructions on generating the required input dataset for Yeti can be found in the supplementary document.





### 3.1.1 HBEFA emission factors and field data

Yeti incorporates HBEFA vehicle subsegment definitions ($k$) into its input and output data representation, from which disaggregated emissions are calculated based on corresponding emission factors. These data are extracted from the desktop version of HBEFA (a Microsoft Access runtime executable) into tab-delimited tabulated text format with corresponding header information. Each emission type (hot / cold excess exhaust, evaporative running losses, evaporative hot soak, evaporative diurnal losses, and non-exhaust PM) is stored in a separate file. Each emission factor file contains ID fields for vehicle subsegment (`IDSubsegment`), category (`IDVehCat`), and pollutant (`IDPollutant`), in addition to emission factor (`EFA`). Additional fields, indicated in Table 2, are also required depending on emission types, consisting of road category (`RoadCat`) and grade (`Grad`), traffic situation (`TS`), as well as cold start and hot soak conditions (`Condition`). Further, the traffic situations field is a concatenation of data on area type (`Area`), road type (`RoadType`), speed limit (`SpeedLimit`), and level of service (`LOS`), all of which requiring additional tables of key / value pairs. Non-exhaust PM emission factors extracted from the desktop version HBEFA 4.1 can be mapped to HBEFA 3.3 vehicle subsegment definitions. Further, corresponding key / value pairs for all ID fields used in the aforementioned emission factor tables are also to be extracted for indexing and cross-referencing in Yeti.

### 3.1.2 Meteorological and seasonal data

Certain emission modes, such as cold excess and all forms of evaporative emissions, are sensitive to meteorological conditions. Furthermore, as oil refineries transition blending for commercial production in a roughly synchronous seasonal basis, systematically altering the evaporative characteristics (RVP) of the fuel for each season. which affects the evaporative emissions through diurnal losses, according to Eqs. (4) and (5). Meteorological conditions correspond to the diurnal ambient temperature profile for the current version of Yeti, while seasonal conditions refer to the fuel RVP and the season (spring, summer, autumn or winter) to which it belongs. Each set of meteorological and seasonal data in Yeti is identified by a unique label and is saved separately in corresponding temperature and seasonal tables.

### 3.1.3 Traffic data and aggregate emissions for the City of Berlin

A composite of source data is used to generate traffic input for Yeti. These sources include HBEFA vehicle subsegment and category distribution, road link topology and properties, vehicle count and HBEFA LOS distribution, compiled in different time periods. Each set of run case traffic data is stored in a root directory. Topological information of the road network is stored in separate tab-delimited tabulated text files under the run case root directory, as with a listing of HBEFA version-specific vehicle subsegments that are present in the traffic scenario. The actual traffic count data can be found in a sub-directory, where traffic flow information for each road link is stored in separate tabulated text files representing each day type, with each tabulated text containing the hourly counts of each vehicle subsegment distributed across all LOSs. The required input data for the City of Berlin can be calculated using the source data presented in Table 3.





Basic topological information of the road network is stored in a shapefile. Each road link can be uniquely accessed by its road segment identification as well as traffic direction, from which attributes such as length ($x^l$) and grade ($\phi$), as well as road type, speed limit, and vehicle capacity, can be obtained. Road segments of zero-length, with no indicated traffic direction, or volume are ignored. In the meantime, the hourly count of each vehicle subsegment under each LOS ($\Lambda_k^l$) can be calculated with Eq. 8 using the available source data:

$$\Lambda_k^l = \varphi_k^\tau \cdot \varphi_\tau^l \cdot \varphi_\lambda^l \cdot N^l, \tag{8}$$

where $\varphi_k^\tau$ is the categorial ($\tau$) fraction of vehicle subsegment ($k$), $\varphi_\tau^l$ is the fraction of vehicle category at road link ($l$) evaluated for the hour, $\varphi_\lambda^l$ is the LOS ($\lambda$) fraction at $l$ for the hour, and $N^l$ is the vehicle count passing through $l$ during over the hour. It should also be pointed out that the highest resolution that can be achieved in the emission inventory is that of the individual road segment, within which vehicular distribution is assumed to be spatially uniform. Additional loss in spatial resolution could also be introduced through the calculation of the traffic flow and corresponding LOS, which is typically derived over a road distance covering at least two intersections. The combination of topological and traffic consideration could impact the highest possible spatial resolution that can be processed by Yeti and other bottom-up emission inventory framework.

The categorial vehicle subsegment fraction ($\varphi_k^\tau$) derived from the annual mean vehicle fleet distribution are available as spreadsheets for the years 2015 and 2020 using license plate recognition at monitoring stations installed in key locations across the city (Schmidt and Düring, 2016; 2021), in which vehicle registration information accompanying the license plates are classified into vehicle subsegments defined in HBEFA. The vehicle subsegment counts are, in turn, normalized by the total vehicle counts under the corresponding HBEFA vehicle categories.

The diurnal fraction of each vehicle category ($\varphi_\tau^l$) is based on counts of each vehicle category passing through the road link of interest and are evaluated under the "daytime" (06h-18h), "evening" (18h-22h), and "night time" (22h-06h) periods. Thus, the hourly categorical distributions for a given road link are assumed uniform within each of the three periods. Accordingly, the product $\varphi_k^\tau \cdot \varphi_\tau^l$ then becomes the fraction of the vehicle subsegment of all vehicles passing through road link ($l$) for the hour.

Furthermore, the total traffic count ($N^l$) and LOS fractions ($\varphi_\lambda^l$) are stored in a tabulated text file, sorted by road segment identification, direction, and local hour. It should be noted that the current traffic count uses LOS definitions from HBEFA 3.3, where only four LOSs are defined. However, a fifth LOS (i.e., Stop-and-Go II) has been introduced into HBEFA 4.1 to represent congested traffic where the mean traffic speed is 10 km h$^{-1}$ or less. Contemplation of possible attribution methods from LOS IV in HBEFA 3.3 to LOS IV and LOS V in HBEFA 4.1 is application specific and is thus beyond the scope of this article. but for the purpose of demonstrating the functionality of Yeti over the two versions of HBEFA, the same traffic data has also been applied to HBEFA 4.1 by attributing all LOS IV traffic from HBEFA 3.3 to HBEFA 4.1.





## 3.2 Comparison with aggregate data

The Berlin data set described in Section 3.1 were used in Yeti to generate aggregate emission levels reported from 2015 (Diegmann et al, 2020). It consists of a total of 10,082 road segments, corresponding to 18,980 direction-specific road links. Annual mean fleet composition data for 2015 and 2020 (Schmidt and Düring, 2016; 2021) were applied to the 2015 traffic count data to generate traffic input data for Yeti specific for HBEFA 3.3 and 4.1, in accordance with the methodology outlined in 3.1.3. The combination of fleet composition year and HBEFA versions gave rise to four possible scenarios. Non-exhaust PM emissions for HBEFA 3.3 were derived from HBEFA 4.1, as described in Section 3.1.1. While vehicle subsegments for the 2020 fleet composition were available for both HBEFA versions, composition for 2015 were mapped from version 3.3 to 4.1 using the methodology described in Appendix A. As a limitation of the source traffic count data, only the first four LOS's have been defined in the originating traffic count data and were processed accordingly for the purpose of operation verification of Yeti under both versions. Mean diurnal temperature profiles for Germany (Fig. 3) as well as seasonal RVP values (Table 4) extracted from HBEFA were used throughout all evaluations.

Computations were performed on a machine with two Intel Xeon Platinum 9242 processors and 384 Gb of physical memory. Preprocessing of traffic data took place a single-core process, requiring about approximately 60 minutes for each HBEFA version. Subsequent Yeti runs were conducted on eight cores taking approximately 140 minutes wall clock time. Aggregation of emission data across all road links and subsegments took another 70 minutes for all pollutants, meteorological profiles, and day types. The postprocessed aggregates were then assembled into annual daily output using a weighted average for each season and day type corresponding to the percentages of each seasonal day type for 2015, which can be found in Table 5.

All annual aggregate emissions based on Yeti are presented in Table 6 as mean daily tonnage with comparison from reported emissions in 2015 from the Berlin Senate (Diegmann et al, 2020), obtained using HBEFA 3.3 under the official aggregated reporting guideline of VDI (2020). The Yeti outputs for CO and HC are at a comparable level with the Berlin senate value of 37.78 tonnes day$^{-1}$. Increased CO and HC emission levels calculated in the 2015 fleet scenario using HBEFA 4.1, at 41.93 tonnes day$^{-1}$ can be attributed to the loss of granularity in passenger vehicle subsegment definitions in the mapping, from 48 vehicle subsegments with non-zero fractions in HBEFA 3.3 to only 16 vehicle subsegments in HBEFA 4.1. This can be rectified by further improvement in mapping methodology. Nitrogen oxides emissions are at a similar level as the Berlin senate value of 15.94 tonnes day$^{-1}$ in both HBEFA versions for the 2015 fleet composition. However, a significant decrease in NOx is observed for the 2020 fleet composition. This is possibly due to the introduction of diesel passenger vehicles with generally lower reported emission factors. Meanwhile, differences in PM emissions can be seen among the Yeti run cases, where results generated using HBEFA 4.1 vehicle subsegment definitions and emission factors are closer to the 1.50 tonnes day$^{-1}$ figure reported by the Berlin Senate, while a significant underprediction appear in both HBEFA 3.3 run cases. This could be explained by the use of HBEFA 4.1 non-exhaust PM emission factors for all run cases, as discussed below, and could be remedied for HBEFA 3.3 by extracting the appropriate emission factors from, for instance, the expert version.





A breakdown of different emission modes for the four Yeti run cases is presented in Table 7. A typical distribution can be seen for all run cases, where the NOx emissions are dominated by hot exhaust emissions, while cold excess emissions make up the majority of CO and HC emissions. Conversely, PM emissions are dominated by non-exhaust contributions. Here the difference in HBEFA versions can be seen, where, as discussed earlier, the use of HBEFA 4.1 non-exhaust factors has caused a noticeable decrease in emission outputs under HBEFA 3.3. On the other hand, the combined contributions from evaporative HC emissions are generally about one magnitude smaller than their counterpart, as indicated by Landman (2001) and USW-EPA (2012), with running losses and hot soak emissions significantly lower than hot and cold excess exhaust emissions. However, an increase in diurnal evaporative emissions to a comparable level with cold excess under HBEFA 4.1 can be observed. This is caused by a general increase in evaporative diurnal emission factor values from version 3.3 to 4.1.

The Yeti run cases with the different fleet composition and HBEFA versions produced aggregate emissions that were similar to the figures reported by the Berlin Senate. Some discrepancies could be observed and could be remedied in part by an improvement of vehicle subsegment mapping methodology between the two HBEFA versions or by explicitly defining vehicle subsegments using the specific version of HBEFA. Moreover, the use of non-exhaust PM emission factors from HBEFA 4.1 on 3.3 vehicle subsegment definitions resulted in a drastic decrease in aggregate PM output. This could be solved by using emission factors derived from the expert version of HBEFA 3.3, where emission factors for non-exhaust PM are available. In addition, an increase in aggregate evaporative increase in evaporative diurnal HC emissions could be attributed to a corresponding increase in emission factors from version 3.3 to 4.1. With these point in mind, the operation of Yeti has been verified under both HBEFA 3.3 and 4.1, where the aggregated emissions have been evaluated against reported values produced in accordance with the procedures set forth by VDI (2020).

## 4 Road link level emissions

In further consideration of the Yeti dataset from Section 3, the emphasis for this section will be placed on hourly emission levels from individual road sections. For the purpose of contrasting different temperature profiles and day types, in a succinct and conclusive manner, the profiles for summer and winter temperature (Figure 3) and accompanying seasonal RVP values (Table 4), in combination with traffic activity profiles representative of workdays (Mondays to Thursdays) and Sundays / holidays are used. Moreover, two road sections – Frankfurter Allee, a trunk road of 3.49 km; and Silbersteinstraße, a collector road of 1.42 km – are featured for the presentation as the road function and correspondingly the traffic flow pattern are anticipated to exert significant influence on the emission output throughout. The diurnal cycle for the pollutant species is illustrated over the two road sections at the different day type and meteorological conditions. This is followed by annual aggregates for the two road sections, distributed over HBEFA vehicle categories. Finally, for concise display of the results, only the pollutants NOx and HC are shown in the sections that follow.





## 4.1 Total Hourly emissions

Figure 4 illustrates the total hourly NOx and HC emissions for typical workday and holiday traffic activities under mean summer and winter diurnal meteorological and seasonal RVP values along Frankfurter Allee and Silbersteinstraße. All emission profiles conform to the traffic pattern expected of the corresponding day type, as evidenced by the characteristic morning and afternoon rush hour peaks for a typical workday, as well as a comparatively steady build-up towards early afternoon on an ordinary holiday. Further, due to the longer length and higher traffic volume of Frankfurter Allee over Silbersteinstraße, the amount of pollutants emitted, especially for NOx, is correspondingly higher on Frankfurter Allee.

For this Yeti emissions data set, trunk roads such as Frankfurter Allee are considered transitory, which implies that no cold-start or hot-soak events are expected. Thus, the summer and winter emission profiles for NOx are identical, since in this case the production mechanism is entirely attributed to hot exhaust emissions. A slight but noticeable difference between the two meteorological conditions (Summer and Winter) can be observed for HC emissions. This is due to dependence of meteorological conditions in evaporative diurnal and running losses emissions, which increases with the ambient temperature. However, the HC emission profiles on Silbersteinstraße are higher in Winter than in Summer, accompanied by a less drastic increase in the NOx emissions. Both can be attributed to contributions from cold excess emissions, which increases in lower ambient temperature, especially in collector roads such as Silbersteinstraße.

## 4.2 Total daily emissions

Tables 8 to 11 show the total daily emissions for workday and holiday traffic activities for summer and winter temperature profiles over the same road sections of interest. These emissions are classified by HBEFA vehicle categories (Tables 8 and 9) and Euro emission standards (Tables 10 and 11). The differences in meteorological profiles observed in Fig. 4, that is, the NOx emissions for Frankfurter Allee are also identical during summer and winter in Table 7; the role of evaporative diurnal and running losses evaporative HC emissions (Table 8) in response to changes in ambient temperature; as well as the contributions from cold excess – and to a lesser extent hot-soak evaporative emissions – on Silbersteinstraße, are also reflected in the total daily emissions data for the same reasons previously mentioned. There are, however, a few noteworthy observations which are further elaborated below. As supplement to the subsequent arguments, the corresponding traffic flow for the two roads have been calculated and classified in Table A1 according to vehicle categories as well as Table A2 according to Euro emissions classes.

First, the contributions from busses on the total NOx emissions on Silbersteinstraße, despite only taking up about 5% of daily traffic volume on the street (Table A1), are consistently at a similar or higher level than those from passenger vehicles presented in Table 8. From a technical perspective, this is indicative of the impact of Euro IV/V compliant selective catalytic reaction technology (SCRT) devices under off-cycle conditions, where NOx conversion efficiency reduces notably due to a lowered exhaust temperature caused by lower-than-demand duty cycles (Lowell and Kamakaté, 2012; Carslaw et al, 2015). The extent of this observation is confirmed by using aggregated HBEFA NOx emission factors detailed in Section A2. This is evidenced





initially in a substantially higher HBEFA aggregate hot exhaust emission factor for NOx – 6.802 g km$^{-1}$ for each bus, compared
to 0.4301 g km$^{-1}$ for each passenger vehicle. Correspondingly, the daily total NOx emissions on Silbersteinstraße calculated
using these aggregated emission factors for passenger vehicles and busses are found to be at a similar level as corresponding
values obtained by Yeti in Table 8, further suggesting the plausibility of this observation. The introduction of Euro VI
compliant SCRT devices in the current bus fleet from 2018, as part of the official air pollutant control strategy (Berlin City
Senate, 2019), should result in a substantial reduction of NOx emissions.

Second, a substantially higher fraction of HC emissions on Frankfurter Allee (Table 9) originate from motorcycles, which only
make up less than 3% of the daily total traffic flow, compared to passenger vehicles, although the air quality standards for
HCs, such as benzene and toluene, have been successfully met (Herenz et al, 2020). In a similar fashion as the NOx emissions
from busses above, a comparison can be made between the contributions of HC emissions from passenger vehicles and
motorcycles using aggregate HBEFA emission factors, in which hot exhaust and diurnal evaporative emissions are dominant.
The aggregate emission factors for each motorcycle are determined to be 1.907 g km$^{-1}$ for hot exhaust and 0.267 g day$^{-1}$ through
diurnal evaporative processes, while they are correspondingly 0.019 g km$^{-1}$ and 0.075 g day$^{-1}$ for each passenger vehicle. This
compensates for the lack of motorcycles, bringing relative contribution of HC emissions to nearly 60% with respect to
passenger vehicles, at a level in line with the figures reported in Table 9. Details of the calculation are presented in Section
A3.

Another interesting observation involves NOx and HC emissions for vehicles belonging to Euro III emission standard and
below, which make up a significant portion of the total contributions, to up to about 70%, for both pollutants, as indicated in
Tables 10 (NOx) and 11 (HC). Based on the data obtained from the license plate recognition study of Schmidt and Düring
(2016), the fraction of vehicles belonging to Euro III is $16.17 \pm 2.77$ %, based on a 95% confidence interval over 7 surveillance
locations inside and outside the low emission zone (Umweltzone) acrcoss Berlin, including Frankfurter Allee. On the other
hand, Euro VI vehicles only represent $7.76 \pm 1.38$ % of the 2015 fleet. The percentages of daily traffic volume for Euro III and
below classes and Euro VI vehicles on Frankfurter Allee and Silbersteinstraße are also consistent with these summary statistics.
The volume, in combination with age as well as antiquated combustion and exhaust aftertreatment technologies associated
with vehicles of this type, thus result in a significant contribution in traffic emissions. In contrast with the fleet composition
data obtained in 2020 (Schmidt and Düring, 2016), the Euro III and below vehicles constitute only $7.67 \pm 0.13$% of the vehicle
fleet in Berlin, while the Euro VI vehicles make up $37.28 \pm 3.49$%, both based on a 95% confidence interval on 10 surveillance
locations. The displacement of Euro III with Euro VI vehicles also explains the reduction of annual daily aggregate NOx
emissions in Table 3 between the 2015 and 2020 fleet compositions.

**4.3 Spatial distribution**

Figure 5 shows the spatial distribution of annual daily mean emissions of NOx and HC over the road network in Berlin. The
emissions for each road segment have been normalized by their respective segment length. In general, for NOx, higher





emissions are found along trunk roads, where overall vehicle counts are higher, while higher HC emissions can be seen in collectors and local roads, in which higher emissions through cold start and hot soak events are expected. The highly non-uniform distribution of the emissions in both cases highlights the importance of using dynamic activity data representative of the region of interest in order to obtain model results, that is, pollutant concentration data, that are relevant to the region and period of interest, as stressed in Kuik et al (2018).

## 5 Summary

Yeti, a computationally scalable traffic emission inventory disaggregation tool based on HBEFA (INFRAS, 1999), has been developed using the Python scripting language for generating hourly vehicular emissions data road level resolutions corresponding to local seasonal and meteorological conditions. Using City of Berlin as a region of interest, hourly street level emission inventories have been produced with diurnal traffic activity data for typical workdays (Monday to Thursday), Fridays, Saturdays, as well as Sundays and holidays in 2015 and 2020 using mean seasonal temperature and RVP profiles. Inventories are produced for main regulated pollutants from hot and cold excess exhaust emissions, as well as non-exhaust contributions, namely evaporation (diurnal, hot soak and running losses) and road and tire abrasion, with HBEFA versions 3.3 and 4.1 using vehicle fleet composition data obtained using license plate recognition for 2015 and 2020 (Schmidt and Düring, 2016; 2021). The emissions of CO, HC, NOx, and PM generated by this tool are found to be consistent with the annual daily aggregate emissions reported by Diegman et al (2020) for the official emission inventory for the City of Berlin.

In addition, the hourly NOx and HC emissions are presented on two road sections with distinctive functions in Berlin – Frankfurter Allee, a trunk road; and Silbersteinstraße, a collector road. The corresponding diurnal emissions profiles show increase in HC emissions, and NOx to a lesser extent, on collector roads in winter due to cold starts, while consistently higher overall emissions can be found on trunk roads resulting from higher traffic volume in comparison, albeit less sensitive to seasonal changes. Daily total emissions show significant to dominant contribution from specific vehicle categories, that is, NOx from busses on collector roads and HC from motorcycles on trunk roads, despite much lower traffic volume than passenger vehicles. On the other hand, a significant fraction of the vehicle fleet belongs to vehicles of Euro III emission standards and below, which represent a single major contributor to traffic emissions, at least on the road sections considered.

Although the current Yeti runs are based on emission factors of HBEFA versions 3.3 and 4.1, it is expected to function with future versions under the same database structure. An immediate interest would be to deploy Yeti in conjunction with the recently released of HBEFA 4.2 (Notter et al, 2022) to inspect its effectiveness. Further, while HBEFA provides excellent information for vehicular emissions mainly for targeted regions in Europe, the applicability of Yeti is also restricted to these regions. It is thus advantageous to consider extending Yeti in the future to work with other road transport emission factor databases, such as the COPERT (Ntziachristos et al, 2009) and MOVES3 (US-EPA, 2021), so that Yeti could be applied to a wider geographical region, in a similar manner as other existing models such as VEIN (Ibarra-Espinosa et al, 2018).





On the other hand, a number of considerations can also be made based on the traffic activity information deployed for the current study. First is the expansion of the number of stop-and-go LOSs from HBEFA 3.3 to 4.1. The splitting criteria from the current single stop-and-go data intended for version 3.3 has not been rigorously explored, but a systematic representation

of this transition, at least for the Berlin traffic data, would provide very useful in its applicability on future versions of HBEFA. Moreover, cold start and hot soak events, as well as their travelling distances are so far estimated in accordance with distributions set forth by VDI (2020) guidelines. Additional observational campaigns could be rendered to obtain information more relevant for the particular study, which could be used in Yeti.





**Appendix A: Street level traffic count data classification and estimation of total daily emissions aggregate HBEFA emission factors**

Based on the vehicle subsegment counts across each road segment for each day type following the methodology outlined in Section 3.1.3, segments belonging to the same road (Frankfurter Allee and Silbersteinstraße) are grouped. Vehicle subsegments counts are summed independently based on category and Euro emission class. The daily count for the entire road is calculated through the weighted mean of the counts over the length of each associated road segment. Table A1 shows the traffic counts for both roads on workdays and Sundays/holidays classified by category, and by Euro emission standards in Table A2.

**A1 Estimating contribution of NOx emissions from busses on Silbersteinstraße**

Equation (1) and all dependent equations are used to calculate total emissions using the aggregated emission factors obtained from HBEFA version 3.3, as presented in Table A3. Contributions to NOx emissions from non-combustion sources are nil and thus not presented. Cold start contribution to NOx emissions are also not expected to be significant and are thus omitted in the estimation.

Using the weekday traffic activity profile as reference, Silberseinstraße has a daily traffic volume of 7464.2 passenger vehicles and 463.6 busses along its length of 1.42 km. The aggregated daily emission from the two categories can then be calculated as the product of the emission factors and the traffic volume over the length of the road. This gives rise to corresponding NOx emissions of 4.56 kg and 4.48 kg for passenger vehicles and busses. Using the same calculation, the NOx emissions for a Sunday/holiday traffic activity profile are found to be 2.73 kg and 2.80 kg respectively. While these values are not (nearly) identical to figures presented in Table 8, they are of comparable magnitude. As such, based on the reported emission factors, it is quite possible that a small number of busses is responsible for such significant contribution to NOx emissions compared to passenger vehicles.

**A2 Estimating contribution of HC emissions from motorcycles on Frankfurter Allee**

The methodology outlined in Section A1 is further applied to inspect the HC contributions from motorcycles on Frankfurter Allee. First, the aggregated emission factors are tabulated. Being a trunk road, the number of cold starts is not expected to be very high, so that are not considered. Also, the contributions from running losses and hot soaks, being two orders of magnitude lower than the hot exhaust emissions, can be effectively regarded as negligible. Therefore, only emission factors for hot exhaust and diurnal evaporation are used, which are listed in Table A4.

There are 53944.2 passenger vehicles and 1577.5 motorcycles passing Frankurter Allee each day according to the weekday traffic activity profile. The hot exhaust emissions for each vehicle category can be calculated as the product of the respective emission factor and vehicle count over the road length of 3.49 km, while the diurnal emissions are obtained by simply multiplying the emission factor in question by the number of applicable vehicles. This yields HC emissions of (3.65 + 4.04) = 7.69 kg for passenger vehicles and (10.50 + 0.42) = 10.92 kg for motorcycles. Repeating the procedure for the Sunday/holiday





traffic activity profiles give HC emissions of (2.08 + 2.31) = 4.39 kg and (5.90 + 0.24) = 6.14 kg respectively. Again, these estimates are comparable with the corresponding values shown in Table 8, which justifies the observed level of HC contribution from motorcycles from Yeti.



## Appendix B: Nomenclature

### B1 Roman symbols

| | |
|---|---|
| $C$ | (Subscript) Cold excess exhaust emission for each cold start event |
| $D$ | (Subscript) Evaporative diurnal emission per vehicle per day |
| $E_n^l$ | Total hourly emission for species $n$ over road link $l$ across all vehicle subsegments [g] |
| $e^l$ | Hourly emission for species $n$ over road link $l$ across all vehicle subsegments [g] |
| $H$ | (Subscript) Hot exhaust emission for each kilometer driven |
| $h$ | Local time [h] |
| $i$ | Generic index |
| $k$ | Index for HBEFA vehicle subsegment |
| $l$ | Road link index |
| $N$ | (Subscript) Non-exhaust emission for each kilometer driven |
| $N^l$ | Hourly count for all vehicles through road link $l$ |
| $N_k^l$ | Hourly count for all vehicles belonging to vehicle subsegment $k$ through road link $l$ |
| $n$ | Emission species index |
| $p_{RV}$ | Reid vapor pressure [kPa] |
| $R$ | (Subscript) Evaporative running loss emission for each kilometer driven |
| $S$ | (Subscript) Evaporative hot soak emission for each engine stop event |
| $T(h)$ | Hourly Ambient temperature over a diurnal cycle [K] |
| $T_{\min}$ | Minimum hourly temperature over a diurnal cycle [K] |
| $\Delta T_a^b$ | Temperature difference for $T(b) - T(a)$ [K] |
| $x^l$ | Length of road link $l$ [km] |

### B2 Greek symbols

| | |
|---|---|
| $\beta$ | Coefficient used in Eq. (4) defined in Table 2 |
| $\varepsilon$ | HBEFA emission factor |
| $\Lambda_k^l$ | Hourly count for all vehicles belonging to vehicle subsegment $k$ through road link $l$ in LOS $\lambda$ |
| $\lambda$ | Index for HBEFA level of service (LOS) |
| $\tau$ | Index for HBEFA vehicle category |
| $\widehat{\Phi}$ | Day to hour emission redistribution factor [day$^{-1}$] |
| $\varphi_k^\tau$ | Fraction of vehicles belonging to vehicle subsegment $k$ in vehicle category $\tau$ [ ] |
| $\varphi_\tau^l$ | Fraction of vehicles belonging to category $\tau$ in the total number of vehicles [ ] |





| $\varphi_\lambda^l$ | Fraction of vehicles belonging to LOS $\lambda$ passing through road link $l$ [ ] |
| $\chi_C^l$ | Fraction of traffic count attributable to cold start events [ ] |
| $\chi_S^l$ | Fraction of traffic count attributable to engine stop events [ ] |

**B3 Acronyms and abbreviations**

| COPERT | Computer Programme to calculate Emissions from Road Transport |
| HBEFA | Handbook Emission Factors for road transport (Handbuch Emissionsfaktoren) |
| HERMES | High-Elective Resolution Modelling Emission System |
| HC | Hydrocarbons (pollutant) |
| LOS | Level of Service |
| MOVES | Motor Vehicle Emission Simulator |
| PM | Particulate matter (specifically $PM_{10}$) |
| SCRT | Selective Catalytic Reaction Technology |
| RVP | Reid vapor pressure ($p_{RV}$) |
| WRF | Weather Research and Forecasting (model) |
| VDI | Society of German Engineers (Verein Deutscher Ingenieure) |
| VEIN | Vehicular Emissions Inventory (R library package) |



*Code and data availability.* The exact version of Yeti described in Section 2.2 are licensed under the terms of the GNU General
Public License version 3.0 or later and can be obtained using the following digital object identifier: 10.5281/zenodo.6594260
(Chan et al, 2022).

*Supplement.* The supplement related to this article is available online at: DOI.

*Author contributions.* The project was originally conceived by TMB and JL. ECC assumed lead design and development. AK
supplied all supporting documentation and evaluation data as well as scientific feedback on the results. The manuscript and all
associated data were prepared by ECC and JL. TMB provided technical guidance at all stages of the study.

*Competing interests.* Some authors are members of the editorial board of journal Geoscientific Model Development. The peer-
review process was guided by an independent editor, and the authors have also no other competing interests to declare.

*Acknowledgements.* All technical concerns on advanced usage of the HBEFA data were craftly addressed by Dr. Jörn Quedenau
(IASS). Mr. Seán Schmitz (IASS) provided valuable feedback on usability and application. A preliminary version of Yeti was
designed and implemented by Mr. Tom Wollnik (Hasso-Plattner-Institut, Potsdam, Germany). The authors also had the great
pleasure of exchanging insights on processing as well as interpretation of traffic emission data with Dr. Annette Rauterberg-
Wulff (Senatsverwaltung für Umwelt, Mobilität, Verbraucher- und Klimaschutz, Berlin, Germany) and Mr. Wim Verhoeve
(EMISIA BVBA, Brussels, Belgium).



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





**Table 1.** Coefficient values for $\beta$ in Eq. (4), adopted from Landman (2001).

| | | | |
|---|---|---|---|
| $\beta_0$ | $8.001\times10^{-3}$ day$^{-1}$ | $\beta_3$ | $-1.944\times10^{-4}$ kPa$^{-1}\cdot$K$^{-2}\cdot$day$^{-1}$ |
| $\beta_1$ | $3.530\times10^{-3}$ K$^{-1}\cdot$day$^{-1}$ | $\beta_4$ | $1.074\times10^{-2}$ K$^{-1}\cdot$day$^{-1}$ |
| $\beta_2$ | $1.733\times10^{-3}$ K$^{-2}\cdot$day$^{-1}$ | $\beta_5$ | $1.008\times10^{-4}$ kPa$^{-1}\cdot$K$^{-1}\cdot$day$^{-1}$ |

**Table 2.** Additional HBEFA data fields required for different emission types.

| Emission type | Yeti name | HBEFA Field(s) |
|---|---|---|
| Hot run | hot | IDGrad, IDTS |
| Cold excess | cold | Condition |
| Diurnal | evapDiurnal | - / - |
| Hot soak | evapSoak | - / - |
| Running losses | evapRL | RoadCat, Condition |
| Non-exhaust PM | nonExhaust | IDGrad, IDTS |

**Table 3.** Source data composite for the city of Berlin for generating Yeti input traffic data.

| File Format | Data Type | Resolution | Year(s) |
|---|---|---|---|
| Spreadsheet | Vehicle subsegment distribution per vehicle category | Annual city mean | 2015, 2020 |
| Shapefile | Road network topology (length, direction, and grade) | Per road link | 2016 |
| | Road link properties (road type, speed limit, and capacity) | Per road link | 2016 |
| | Vehicle category distribution | Per road link, hourly* | 2014 |
| Tabulated text | Total vehicle count | Per road link, hourly | 2015 |
| | Total vehicle LOS distributions for all day types | Per road link, hourly | 2015 |

*Hourly data are collected in the following periods: 06h-18h, 18h-22h, 22h-06h.

**Table 4.** HBEFA seasonal values for fuel Reid vapor pressure for Germany.

| Season | RVP [kPa] |
|---|---|
| Spring | 65.4 |
| Summer | 58.2 |
| Fall (Autumn) | 71.1 |
| Winter | 85.2 |

**Table 5.** Percentages of seasonal day types for 2015 (W: Workdays, F: Fridays, S: Saturdays, N: Sundays and Holidays) used for the calculation of annual aggregate emission outputs.

| Season | W | F | S | N |
|---|---|---|---|---|
| Spring (Mar-May) | 13.42 | 3.29 | 3.56 | 4.93 |
| Summer (Jun-Aug) | 14.52 | 3.56 | 3.56 | 3.56 |
| Fall (Sep-Nov) | 14.25 | 3.56 | 3.29 | 3.84 |
| Winter (Dec-Feb) | 13.97 | 3.29 | 3.29 | 4.11 |





**Table 6.** Comparison of Yeti annual daily aggregate emissions [tonnes day$^{-1}$] with reported values.

| Emissions | CO | HC | NOx | PM |
|---|---|---|---|---|
| Berlin Senate 2015 | 37.78 | 6.78 | 15.94 | 1.50 |
| Yeti (Fleet composition year / HBEFA version) | | | | |
| 2015 / 3.3 | 33.83 | 7.60 | 15.25 | 0.97 |
| 2015 / 4.1* | 41.93 | 13.49 | 14.48 | 1.77 |
| 2020 / 3.3 | 32.65 | 6.45 | 8.26 | 0.76 |
| 2020 / 4.1 | 30.10 | 9.15 | 9.57 | 1.53 |

*Vehicle subsegment definitions from the 2015 fleet were mapped from HBEFA version 3.3 to 4.1.

**Table 7.** Breakdown of aggregate emissions [tonnes day$^{-1}$] by emission type for each Yeti run case (fleet composition year / HBEFA version).

| 2015 / 3.3 | CO | HC | NOx | PM |
|---|---|---|---|---|
| Hot run | 11.18 | 1.80 | 14.72 | 0.21 |
| Cold excess | 22.65 | 5.03 | 0.53 | 0.03 |
| Diurnal | - - | 0.62 | - - | - - |
| Hot soak | - - | 0.04 | - - | - - |
| Running losses | - - | 0.11 | - - | - - |
| Non-exhaust PM | - - | - - | - - | 0.74 |
| **2015 / 4.1** | **CO** | **HC** | **NOx** | **PM** |
| Hot run | 14.50 | 1.78 | 13.95 | 0.33 |
| Cold excess | 27.43 | 5.98 | 0.53 | 0.03 |
| Diurnal | - - | 5.59 | - - | - - |
| Hot soak | - - | 0.00 | - - | - - |
| Running losses | - - | 0.14 | - - | - - |
| Non-exhaust PM | - - | - - | - - | 1.40 |
| **2020 / 3.3** | **CO** | **HC** | **NOx** | **PM** |
| Hot run | 10.46 | 0.83 | 7.62 | 0.09 |
| Cold excess | 22.18 | 4.73 | 0.64 | 0.01 |
| Diurnal | - - | 0.69 | - - | - - |
| Hot soak | - - | 0.06 | - - | - - |
| Running losses | - - | 0.15 | - - | - - |
| Non-exhaust PM | - - | - - | - - | 0.65 |
| **2020 / 4.1** | **CO** | **HC** | **NOx** | **PM** |
| Hot run | 11.35 | 0.83 | 8.93 | 0.11 |
| Cold excess | 18.74 | 3.73 | 0.64 | 0.01 |
| Diurnal | - - | 4.47 | - - | - - |
| Hot soak | - - | 0.00 | - - | - - |
| Running losses | - - | 0.14 | - - | - - |
| Non-exhaust PM | - - | - - | - - | 1.40 |





**Table 8.** Total daily NOx emissions [kg] for typical workdays and holidays from Yeti using mean summer and winter temperature on Frankfurter Allee and Silbersteinstraße in Berlin, classified by HBEFA vehicle categories.

| | HBEFA Category | Workdays | | | | Holidays | | | |
| | | Summer | | Winter | | Summer | | Winter | |
|---|---|---|---|---|---|---|---|---|---|
| **Frankfurter Allee** | Passenger vehicles | 62.78 | (65.1%) | 62.78 | (65.1%) | 33.65 | (66.4%) | 33.65 | (66.4%) |
| | Light commercial vehicles | 4.33 | (4.5%) | 4.33 | (4.5%) | 2.35 | (4.6%) | 2.35 | (4.6%) |
| | Heavy goods vehicles | 19.24 | (19.9%) | 19.24 | (19.9%) | 8.94 | (17.6%) | 8.94 | (17.6%) |
| | Busses | 4.87 | (5.0%) | 4.87 | (5.0%) | 3.37 | (6.6%) | 3.37 | (6.6%) |
| | Coaches | 4.71 | (4.9%) | 4.71 | (4.9%) | 2.07 | (4.1%) | 2.07 | (4.1%) |
| | Motorcycles | 0.54 | (0.6%) | 0.54 | (0.6%) | 0.33 | (0.7%) | 0.33 | (0.7%) |
| **Silberstein-Straße** | Passenger vehicles | 3.56 | (34.4%) | 4.74 | (41.0%) | 2.05 | (33.6%) | 2.74 | (40.2%) |
| | Light commercial vehicles | 0.12 | (1.2%) | 0.18 | (1.5%) | 0.07 | (1.1%) | 0.10 | (1.5%) |
| | Heavy goods vehicles | 0.76 | (7.4%) | 0.76 | (6.6%) | 0.43 | (7.1%) | 0.43 | (6.3%) |
| | Busses | 5.41 | (52.3%) | 5.41 | (46.7%) | 3.28 | (53.8%) | 3.28 | (48.0%) |
| | Coaches | 0.45 | (4.4%) | 0.45 | (3.9%) | 0.25 | (4.1%) | 0.25 | (3.7%) |
| | Motorcycles | 0.03 | (0.3%) | 0.03 | (0.3%) | 0.02 | (0.3%) | 0.02 | (0.3%) |

**Table 9.** Total daily HC emissions [kg] for typical workdays and holidays from Yeti using mean summer and winter temperature profiles on Frankfurter Allee in Berlin, classified by HBEFA vehicle categories.

| | HBEFA Category | Workdays | | | | Holidays | | | |
| | | Summer | | Winter | | Summer | | Winter | |
|---|---|---|---|---|---|---|---|---|---|
| **Frankfurter Allee** | Passenger vehicles | 9.33 | (40.7%) | 7.37 | (35.2%) | 5.56 | (41.9%) | 4.48 | (36.8%) |
| | Light commercial vehicles | 0.07 | (0.3%) | 0.07 | (0.3%) | 0.03 | (0.3%) | 0.03 | (0.3%) |
| | Heavy goods vehicles | 0.49 | (2.1%) | 0.49 | (2.3%) | 0.23 | (1.7%) | 0.23 | (1.9%) |
| | Busses | 0.06 | (0.3%) | 0.06 | (0.3%) | 0.04 | (0.3%) | 0.04 | (0.3%) |
| | Coaches | 0.12 | (0.5%) | 0.12 | (0.6%) | 0.05 | (0.4%) | 0.05 | (0.4%) |
| | Motorcycles | 12.84 | (56.1%) | 12.84 | (61.3%) | 7.34 | (55.4%) | 7.34 | (60.3%) |
| **Silberstein-Straße** | Passenger vehicles | 4.50 | (83.2%) | 9.55 | (91.2%) | 2.61 | (83.1%) | 5.589 | (91.2%) |
| | Light commercial vehicles | 0.035 | (0.6%) | 0.043 | (0.4%) | 0.020 | (0.6%) | 0.025 | (0.4%) |
| | Heavy goods vehicles | 0.019 | (0.4%) | 0.019 | (0.2%) | 0.011 | (0.3%) | 0.011 | (0.2%) |
| | Busses | 0.059 | (1.1%) | 0.059 | (0.6%) | 0.046 | (1.2%) | 0.036 | (0.6%) |
| | Coaches | 0.012 | (0.2%) | 0.012 | (0.1%) | 0.007 | (0.2%) | 0.007 | (0.1%) |
| | Motorcycles | 0.79 | (14.5%) | 0.79 | (7.5%) | 0.46 | (14.6%) | 0.459 | (7.5%) |





**Table 10.** Total daily NOx emissions [kg] for typical workdays and holidays from Yeti using mean summer and winter temperature profiles on Frankfurter Allee and Silbersteinstraße in Berlin, classified by Euro emission standards.

|  | | **Workdays** | | | | **Holidays** | | | |
|---|---|---|---|---|---|---|---|---|---|
|  | **Euro Emission Standard** | **Summer** | | **Winter** | | **Summer** | | **Winter** | |
| **F-Allee** | Euro III and below | 17.85 | (18.5%) | 17.85 | (18.5%) | 9.69 | (19.1%) | 9.69 | (19.1%) |
|  | Euro IV | 21.78 | (22.6%) | 21.78 | (22.6%) | 11.42 | (22.5%) | 11.42 | (22.5%) |
|  | Euro V | 52.34 | (54.3%) | 52.34 | (54.3%) | 27.19 | (53.6%) | 27.19 | (53.6%) |
|  | Euro VI | 4.48 | (4.6%) | 4.48 | (4.6%) | 2.41 | (4.7%) | 2.41 | (4.7%) |
| **S-Str** | Euro III and below | 3.64 | (35.2%) | 3.78 | (32.7%) | 2.19 | (35.8%) | 2.27 | (33.2%) |
|  | Euro IV | 2.20 | (21.3%) | 2.61 | (22.6%) | 1.30 | (21.3%) | 1.54 | (22.6%) |
|  | Euro V | 4.24 | (41.0%) | 4.87 | (42.1%) | 2.47 | (40.5%) | 2.84 | (41.6%) |
|  | Euro VI | 0.25 | (2.5%) | 0.31 | (2.7%) | 0.15 | (2.4%) | 0.18 | (2.6%) |

**Table 11.** Total daily HC emissions [kg] for typical workdays and holidays from Yeti using mean summer and winter temperature profiles on Frankfurter Allee and Silbersteinstraße in Berlin, classified by Euro emission standards.

|  | | **Workdays** | | | | **Holidays** | | | |
|---|---|---|---|---|---|---|---|---|---|
|  | **Euro Emission Standard** | **Summer** | | **Winter** | | **Summer** | | **Winter** | |
| **F-Allee** | Euro III and below | 15.75 | (68.8%) | 15.34 | (73.2%) | 8.97 | (67.7%) | 8.74 | (71.8%) |
|  | Euro IV | 3.81 | (16.7%) | 2.93 | (14.0%) | 2.31 | (17.4%) | 1.82 | (15.0%) |
|  | Euro V | 2.76 | (12.0%) | 2.22 | (10.6%) | 1.63 | (12.3%) | 1.33 | (10.9%) |
|  | Euro VI | 0.57 | (2.5%) | 0.46 | (2.2%) | 0.34 | (2.6%) | 0.28 | (2.3%) |
| **S-Str** | Euro III and below | 2.04 | (37.6%) | 3.30 | (31.5%) | 1.18 | (37.6%) | 1.93 | (31.5%) |
|  | Euro IV | 1.93 | (35.6%) | 4.25 | (40.6%) | 1.12 | (35.5%) | 2.49 | (40.6%) |
|  | Euro V | 1.22 | (22.5%) | 2.46 | (23.5%) | 0.71 | (22.6%) | 1.44 | (23.6%) |
|  | Euro VI | 0.23 | (4.3%) | 0.45 | (4.4%) | 0.13 | (4.3%) | 0.27 | (4.3%) |





**Table A1.** Mean daily traffic count for typical workdays and holidays on Frankfurter Allee and Silbersteinstraße in Berlin, classified by HBEFA vehicle categories.

| HBEFA Category | Frankfurter Allee | | | | Silbersteinstraße | | | |
|---|---|---|---|---|---|---|---|---|
| | Workdays | | Holidays | | Workdays | | Holidays | |
| Passenger vehicles | 53944.2 | (90.7%) | 30779.6 | (90.7%) | 7464.2 | (87.1%) | 4473.1 | (87.0%) |
| Light commercial vehicles | 2024.6 | (3.4%) | 1138.2 | (3.4%) | 206.1 | (2.4%) | 121.4 | (2.4%) |
| Heavy goods vehicles | 1555.0 | (2.6%) | 873.9 | (2.6%) | 157.5 | (1.8%) | 92.6 | (1.8%) |
| Busses | 203.9 | (0.3%) | 149.0 | (0.4%) | 463.6 | (5.4%) | 289.5 | (5.6%) |
| Coaches | 162.3 | (0.3%) | 90.8 | (0.3%) | 40.7 | (0.5%) | 23.8 | (0.5%) |
| Motorcycles | 1577.5 | (2.7%) | 886.7 | (2.6%) | 238.9 | (2.8%) | 140.4 | (2.7%) |

**Table A2.** Mean daily traffic count for typical workdays and holidays on Frankfurter Allee and Silbersteinstraße in Berlin, classified by Euro emission standards.

| Euro Emission Standard | Frankfurter Allee | | | | Silbersteinstraße | | | |
|---|---|---|---|---|---|---|---|---|
| | Workdays | | Holidays | | Workdays | | Holidays | |
| Euro III and below | 10901.4 | (18.3%) | 6540.0 | (18.3%) | 1649.1 | (19.1%) | 988.5 | (19.2%) |
| Euro IV | 22420.4 | (37.7%) | 13455.8 | (37.7%) | 3140.5 | (36.6%) | 1882.8 | (36.6%) |
| Euro V | 21717.9 | (36.5%) | 13037.5 | (36.5%) | 3139.4 | (36.6%) | 1884.2 | (36.7%) |
| Euro VI | 4427.8 | (7.5%) | 2658.5 | (7.5%) | 642.1 | (7.5%) | 385.4 | (7.5%) |

**Table A3.** Aggregated NOx emission factors for busses and passenger vehicles for HBEFA 3.3 for Germany.

| NOx emission factor | Pass. Veh. | Bus |
|---|---|---|
| $\varepsilon_H [\text{g km}^{-1}]$ | 0.4301 | 6.802 |
| $\varepsilon_C [\text{g}]$ | 0.0716 | N/A |

**Table A4.** Aggregated HC emission factors for busses and passenger vehicles for HBEFA 3.3 for Germany.

| HC emission factor | Pass. Veh. | Motorcycle |
|---|---|---|
| $\varepsilon_H [\text{g km}^{-1}]$ | 0.0194 | 1.907 |
| $\varepsilon_D [\text{g day}^{-1}]$ | 0.0749 | 0.267 |



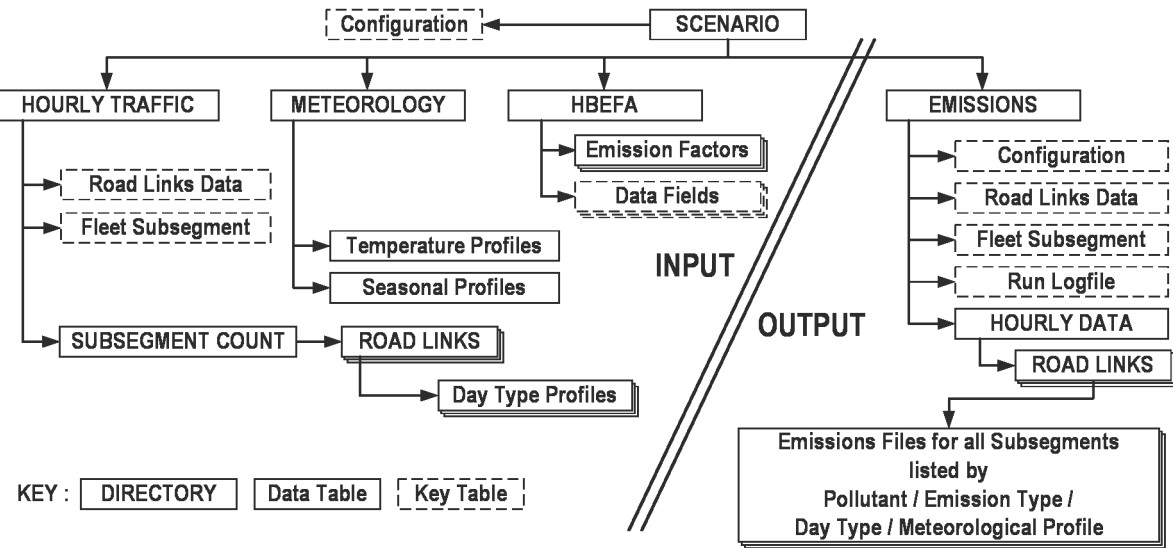

**Figure 1.** Data requirements and organization of Yeti.





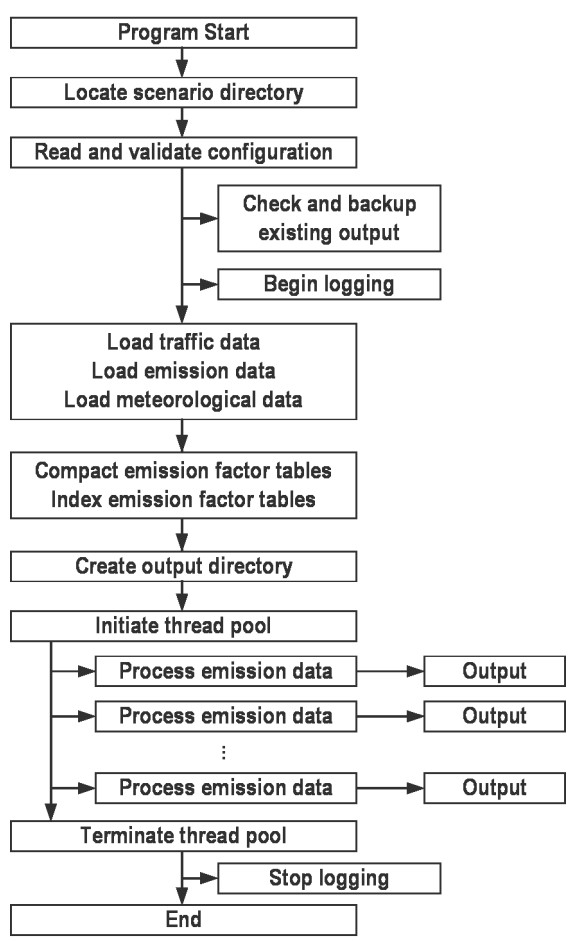

**Figure 2.** Execution flow of Yeti.





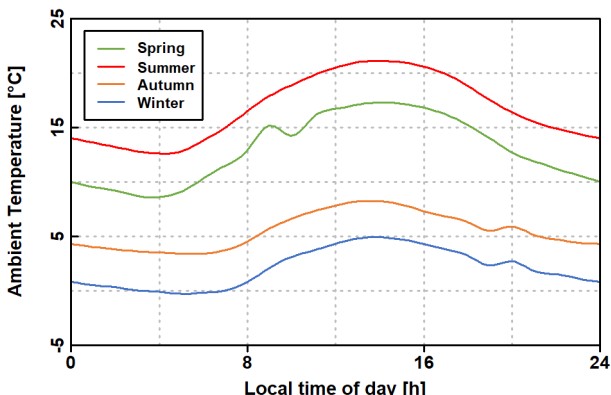

**Figure 3.** HBEFA mean seasonal diurnal temperature profiles for Germany.

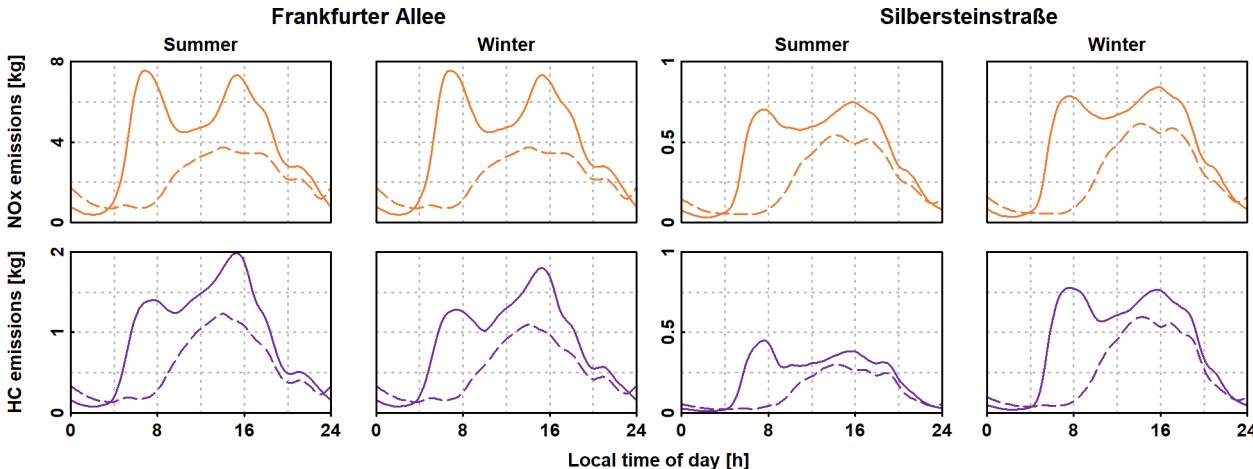

**Figure 4.** Diurnal emissions profiles for NOx (top row) HC (bottom row) from Yeti using mean summer and winter temperature profiles (Fig. 3 and Table 4) on Frankfurter Allee and Silbersteinstraße in Berlin. Solid lines indicate traffic activity profile for a typical weekday, and dashed lines indicate a typical holiday traffic activity profile.

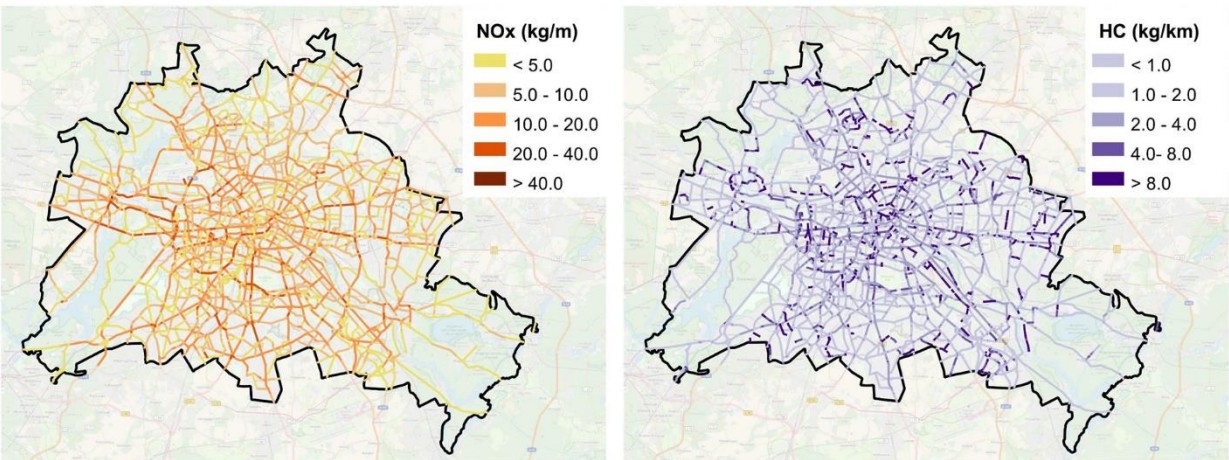

**Figure 5.** Distribution of annual daily mean (left) NOx and (right) HC emissions [kg m$^{-1}$] for 2015 from Yeti using HBEFA 3.3 emission factors over the Berlin road network (Background overlay © Google Maps 2022).

740