# Peer review of "Yeti 1.0: a generalized framework for constructing bottom-up emission inventory from traffic sources at road link resolution"

_Geoscientific Model Development, 2022_

## Author Comment (AC1)

Dear Editor:

Please find the following pages our response to the three RCs for the present MS. Responses to the individual RC has also been sent to the Reviewer responsible. An ATC of the MS is also provided for review. We would like to this opportunity to summarize below a few main issues.

First, Reviewer 2 suggested that the MS should indicate the target audience more directly. In this spirit the title of the MS has been changed (slightly) to "Yeti 1.0: a generalized framework for constructing bottom-up emission inventory from traffic sources at road link resolution". Further remarks are also introduced to emphasize the use of Yeti for the purpose of generate traffic emissions for use with urban and regional scale air quality models.

Second, there are concerns from all three Reviewers of varying degrees on the accuracy of the input traffic activity and emission factors used in demonstrating the functionality of Yeti. The present MS has been submitted to GMD as a Model Description Paper, and as such the focus is on the methodology in which the input data – namely, traffic activity, meteorological data, and HBEFA emission factors – can be used by Yeti to generate emission data that are consistent with existing emission inventory (§3.2) and road side emissions that are consistent in the input (§4). In the view of the Authors, a rigorous scrutiny of all source data, while requisite for a model evaluation article, is out of the scope for a paper dedicated to methodology or model description. Therefore, here the manuscript has been limited to the main goal of simply presenting and describing Yeti as a tool, avoiding purposefully the further analysis of data used. The same argument also applies to the observation from Reviewer 2 and 3, that the MS reads more as an extended user manual than a scientific article, where descriptions on architecture and usage are considered integral components of a GMD model description paper.

Third, the treatment of cold excess and, to a lesser extent, evaporative hot soak emissions – in particular, the assignment of a fraction of road traffic based on certain road types to cold start and engine stop events, and their arbitrary choice for values – have been brought to question. This practice is in line with an existing work using HBEFA emission factors, where the reference has been provided to the responses and in the MS. In the absence of such information in the input traffic activity data these fractions could only be arbitrarily selected, §§2.1.2, 2.1.3, and 5 of the MS has been amended to emphasize the configurability of said fractions inside Yeti, without code change, should information on cold start and engine stop events be available to the user.

In light of the above, all of the Reviewers' concerns have been fully addressed to the Authors' best ability and understanding of them.

Sincerely,

The Authors.

**Response to Reviewer 1**

The Authors would like to thank the Reviewer for an open and transparent critique on the MS. Please find our response on the corresponding comments below, with reference to the original text in italics, with editions indicated in brackets to reflect the Authors' interpretation of said comments.

**Comment 1:** *… there [is] no explanation [on] basic concepts such as traffic situation …*

**Response:** Section 2.1.1 has been amended to provide an explanation of traffic situation.

**Comment 2:** *… or mentioning the PHEM model, which apparently generates the emission factors inside HBEFA.*

**Response:** While HBEFA uses PHEM to generate hot run emission factors, Yeti does not work directly with PHEM emission factors and thus a discussion on the PHEM model is beyond the scope of this MS.

**Comment 3:** *The manuscript also [lacks] comparison with other estimates such as the % of cold start and evaporative [hot soak] emissions with other areas or cities. The expectation is that different estimates will converge to similar results, rather than produce equal values.*

**Response:** The comparison of emissions, or specific types of emissions with other areas or cities is more suitable for a model evaluation paper. As the MS is submitted as a model description paper, it suffices to demonstrate cold start and evaporative hot soak emissions can be account for in a manner that adequately reflect the input traffic activity. This is further substantiated by similar works, where Ibarra-Espinosa et al (2018) and Gavidia-Calderón et al (2021) focused exclusively on São Paulo (Brazil), Verrati et al (2020) performed their study on Modena (Italy), Perugu et al (2017) featured Cincinnati (Ohio, USA) in their work, Gurney et al (2017) used Indianapolis City (Indiana, USA) as their region of interest, while Singh et al. (2020) looked at the City of London. That the Authors featured the city of Berlin in the present MS is, by inference, in-line with existing expectations and practices.

**Comment 4:** *Long and ambiguous phrases on lines (lines 34-37, 54-58 and more).*

**Response:** The sentences in this MS are as long as they are necessary to completely and accurately convey the various complex concepts at hand. Further, the Reviewer's claim on the phrases being ambiguous requires further substantiation (i.e., what, how, and why they are ambiguous) through which concrete, constructive and actionable responses can be rendered, especially when they are identified as being "major" issues.

**Comment 5:** *Also, there are paragraphs consisting [of] only two sentences. Each paragraph should have at least three sentences.*

**Response:** Rudimentary heuristics on paragraph composition based solely on length and form – exemplified by the three-sentence or five-sentence rule – ignore the role of the paragraph as a unit of thought (Cutts, 2020, p 29) and a device of punctuation (Clark, 2002, p 129), where it can be "a single, short sentence or a passage of great duration" (Strunk and White, 2011, §13), a notion further echoed by Cutts (2020, p 172), who asserted that a paragraph "can be a single sentence, whether long, short, or middling."

**Comment 6:** *Line 80. The authors mention that models WRF Chem and OpenFOAM [] could be integrated with emissions models. Nevertheless, I would recommend the author to also mention the models MUNICH and CityChem, which have methods to estimate the dispersion of street emissions.*

**Response:** After some contemplation, the Authors have decided to refrain from mentioning integrating Yeti with the models MUNICH (Gavidia-Calderón et al, 2021) and CityChem (Ramacher and Karl, 2020) at the risk of misrepresentation. The Authors do not have sufficient technical knowledge of either model to ascertain such possibility. This could, however, be an excellent starting point for studies beyond the scope

of this MS should a future collaboration with the Authors using Yeti, in conjunction with bespoke models or other models, be entertained.

**Comment 7:** *[Section] 2[:] I do not see [a] definition [for] traffic situation.*

**Response:** Please refer to Authors' response to Comment 1

**Comment 8:** *The definition of the emission factors [such] as [the] "epsilon" symbols is not defined.*

**Response:** The symbols for the HBEFA emission factors (the "epsilon" symbols) are defined in line 98 of the MS, and this definition appears again in the Nomenclature section in Appendix B2. The Authors do not understand the purpose of needing to define the definition of emission factors, but the role of the HBEFA emission factors under each emission mode (i.e., hot run, cold excess, evaporative, and non-exhaust) is described in §§2.1.1 to 2.1.4, as well as expressed in Equations (2,3,5,6,7,8), in the MS.

**Comment 9:** *Regarding equation 2, it is not clear for me [what] the temporal resolution of the data [is]. Later, the author mention that the traffic counts represent certain hourly intervals during the day. Then, what are the units of the emissions of equation 2.*

**Response:** Section 2.1 is entitled "Hourly emission calculation strategies" in the MS, which implies Equation (2), being presented in Section 2.1.1, as well as with other equations shown under §2, is expressed on an hourly basis. This will be emphasized throughout §2.1.

**Comment 10:** *[The] [c]old start emissions[] in equation 3 have [a] parameter N which represents hourly traffic counts. Is N different with the parameter in Eq 2 that represent traffic counts?*

**Response:** Since there is no $N$ term in Equation (2), the Authors assume the Reviewer is referring to the expression on line 108 in the MS, where the symbol $N$, and its relationship with the $\Lambda_k^l$ term, is defined. The definition of symbol $N$ appears again in the Nomenclature section in Appendix B2. In the unlikely event of the $N$ term represented different parameters between Equations (2) and (3) an no alternative symbol could be found to distinguish the terms, a change in definition would have been explicitly indicated in the text and Nomenclature.

**Comment 11:** *The methodology for cold starts expressed in Equation 2[] assumes that these emissions only occur[] in collectors and access roads. I understand that the authors are assuming that most of cold start emissions are in these type[s] of roads, which are associated with higher residential areas where there are more cars. If this is the [reasoning], I think it is plausible and reasonable considering the usual limited input data when performing emissions inventories. However, this should be stated explicitly, then, future estimates can improve the current limitation. For instance, a[] possible future improvement could be associate type of roads with residential density and land use. In conclusion, authors need to be explicit and direct.*

**Response:** The value of the $\chi$ terms in Equations (3) and (6) can be changed. Lines 126 to 128 of the MS states that "[f]or the current study, the value of $\chi_C^l$ has been set to 0.3 for all collectors and access roads … and zero for all other road types", implying an application-specific configuration. In addition, the last sentence of the MS also directly and explicitly stated the need to measure cold start and hot soak events so that the corresponding emissions are better represented.

Sections 2.1.2 and 2.1.3 have been amended to stress the configurability of the $\chi$ terms.

**Comment 12:** *"fuels and various volatile fluids", what are [these] various volatile fluids?*

**Response:** The phrase "and various volatile fluids" will be removed from the MS.

**Commen13:** *Running Losses evaporative emissions, equation [6], parameter [$\chi_S^l$]. Are we assuming the value of 0.3 for [$\chi_S^l$]?*

**Response:** Yes, $\chi_S^l$ in equation 7 for evaporative hot soak emissions assumes the same value as the $\chi_C^l$ in equation 3 for cold excess emissions and §2.1.5 has been amended accordingly.

**Comment 14:** *Equation 8. As the non-exhaust emission factors represent both[] [resuspension] and wear emissions, it still not clear form me, [whether it is] the sum or the average?*

**Response:** Lines 164 to 166 of the MS indicate that "HBEFA provides a simplified form for calculating non-exhaust particulate matter, in which estimates for emission factors are available without distinguishing between contributions from abrasion and resuspension," meaning that it is a sum. Also, the concept of an "average" emission factor is meaningless at a vehicle level, as aggregation is necessary to obtain the total emission over the region of interest before any statistical analysis can be performed.

**Comment 15:** *Line 272, what is categorical? The type of vehicle? Or a fraction that is associated with a vehicle category?*

**Response:** The phrase "categorical ($\tau$) fraction of vehicle subsegment ($k$)" will be rewritten to "fraction of vehicle subsegment ($\tau$) belonging to each category ($k$)."

**Comment 16:** *Section 3.[1.]3, do we have a different vehicular composition by hour?*

**Response:** Yes. The discussion on temporal variation of the vehicular composition can be found in §3.1.3, in lines 284-287 of the MS, in which "[t]he diurnal fraction of each vehicle category [symbol] is based on counts of each vehicle category passing through the road link of interest and are evaluated under the "daytime" (06h-18h), "evening" (18h-22h), and "night time" (22h-06h) periods. Thus, the hourly vehicle categorical distributions for a given road link are assumed uniform within each of the three periods. Accordingly, the product [symbol] then becomes the fraction of the vehicle subsegment of all vehicles passing through road link [symbol] for the hour."

**Comment 17:** *Where are []level of services (LOS) [defined]?*

**Response:** A listing of LOSs according to HBEFA definitions will be defined in 2.1.1

**Comment 18:** *What is the []purpose of comparing emission factors from HBEFA 3.3 and 4.1?*

**Response:** No comparison of emission factors between HBEFA 3.3 and 4.1 was made in the MS, as they are being used unmodified as input data for Yeti. However, the purpose of deploying Yeti with both HBEFA 3.3 and HBEFA 4.1 is (1) to demonstrate that the Yeti functions under both versions and, by extension, any possible future versions, and (2) to highlight any necessary considerations in using the two version versions, particularly the need to reconcile subsegment definitions should the traffic activity data be defined using a different version of HBEFA, a situation of regular occurrence. These points are discussed in §3.2 of the MS, with further details to be found in the supplementary document.

**Comment 19:** *Table 6, It is better to [use] totals rather than averages. This is because the spatial distribution of traffic flow and emissions [are] not normal[ly distributed]. There are few roads with most of emissions[. ]It is also necessary to compare emission factors.*

**Response:** The unit presented in Table 6 is presented in the annual daily average aggregated over the entire road network. Thus, the traffic emissions are only valid at a city level and spatial distribution. The Authors also fail to understand the importance of comparing emission factors. In particular, what type of emission factor does the Reviewer have in mind? If it is the typical "output per unit road segment" type of emission factor, then they have been presented in Figure 5 and in the accompanying discussion in §4.3.

**Comment 20:** *Line 321, t[he] lower NOx emissions for 2020: "This is possibl[y] due to the introduction of diesel passenger vehicles with generally lower reported emission factors". This statement implies that we need to a comparison of the emission factors is needed. Furthermore, Carslaw et al (2011) found that diesel vehicles with newer emissions standards emit higher emissions on real world. We also have dieselgate. Then again, a comparison of the emission factors is needed.*

**Response:** As Yeti only employs HBEFA emission factors verbatim and does not recalculate or modify them, further comparison, or discussion on the validity, of HBEFA emission factors is *ultra vires*, especially for a MS submitted as a model description paper, where the focus is to demonstrate that Yeti can produce NOx and other emissions appropriate for the particular version of HBEFA. Here, the Authors stress that Yeti calculates emissions from HBEFA emission factors unmanipulated, and does not generate its own emission factors. However, the Authors would like to point out that the updates in HBEFA during the diesel scandal period were, in part, to correct the emission factors to account for manipulated diesel vehicles (from HBEFA 3.2 to 3.3) and the subsequent hardware and software mitigation (from HBEFA 3.3 to 4.1).

**Comment 21:** *How important are cold start and [evaporative] emissions[?]*

**Response:** The contributions of all emission modes are summarized in Table 7, and are discussed in §3.2, on the paragraph starting on line 327, in the MS, stating that "cold excess emissions make up the majority of CO and HC emissions" and "the combined contributions from evaporative HC emissions are generally about one magnitude smaller than their counterpart".

**Comment 22:** *Is it possible to include other methodologies in Yeti rather than solely depending on HBEFA? For instance, include emission factors and functions to consider wear PM emissions separated from [resuspension]?*

**Response:** While this point has not be explicitly indicated, the need to include COPERT emission factors are mentioned in §5 as a possible area of extension.

**Comment 23:** *I think it would be better if [Figures] 3 and 4 [are] represented [for] the whole city [rather] than [on] specific roads[. ]Lines 390-394, as we can expect different vehicular [emissions?] by [the] type of road, I do not think it is [appropriate] to compare emissions based on few roads.*

**Response:** Aggregate emissions are discussed in §3.2, while spatial distribution of city-wide emissions, at least for $NO_x$ and HC, are presented in §4.3 and Figure 5 of the MS. The objective of §§4.1 and 4.2, and thus Figures 3 and 4, is to demonstrate the ability of Yeti for generating emissions for specific road segments at a level in which detailed observations can be made. As Yeti is to be used in conjunction with regional and urban scale modelling runs, the Authors find the discussion on traffic emissions at road segment resolution highly appropriate.

**Comment 24:** *Lines 361-363, I think it is necessary to include a plot of the volumes for the city, with an appropriate color scheme to [easily] identify the roads with highest volume.*

**Response:** The focus of the present MS is the calculation of emission levels using traffic activity data that are provided to the Authors. There are observations in §4.2, where the volume of traffic as well as inspection of the relevant emission factors are required is necessary to provide an adequate explanation. Any additional illustration or discussion on traffic volume in the city that are used as input to Yeti, are beyond the scope of a model description paper.

**Comment 25:** *Lines 400-404, what are [aggregated] emission factors?*

**Response:** The term "aggregated emission factors" has been amended to "annually aggregated emission factors."

**Comment 26:** *Lines 414-415, which pollutant?*

**Response:** The figures presented in lines 414-415 refer to the percentages of vehicles belonging to the relevant Euro emission standards as presented in Tables 10 and 11, not to any pollutant.

**Comment 27:** *Line 420, what is the purpose of providing normalized street emissions (total by road / length of road)? I strongly believe it should be better to inform simply the totals?*

**Response:** The emissions presented in Figure 5 are originally obtained in vector form, that is, over each road segment. They must be displayed in normalized form (i.e., g per m road section) so that the emission levels are not confounded by the length of the individual road segments.

**Comment 28:** *Identify the areas with more surface emission fluxes is important. This can be done by gridding the street emissions into a spatial polygon grid. The process must be mass conservative, [meaning] that the emissions inside each pixel, should be the same as the emissions of the road inside that pixel. Here the emissions can be represented as fluxes (mass / area / time) or as emissions maps (mass / time). For reference, the EDGAR emission data can be obtained as emission fluxes (NetCDF) or mass/time (.txt). A standard recommendation for emission fluxes is mass/ hour / km2 with spatial distancing of 1 km.*

**Response:** The scope of Yeti concerns the generation of traffic emissions at a road segment level. While a brief description of a mass-conservative gridding process has been included in the supplementary document. The Authors plan to generalize this gridded method for unstructured grid models such as OpenFOAM, which forms the basis for another dedicated journal article.

**References**

Clark, MC (2002) A Matter of Style. Oxford University Press.

Cutts M (2020) Oxford Guide to Plain English 5 Ed. Oxford University Press.

Gavidia-Calderón ME, Ibarra-Espinosa S et al (2021) Geosci Model Dev 14(6) 3251-3268.

Gurney K et al (2017) Elem Sci Anth 5 44.

Ibarra-Espinosa S, et al (2018) Geosci Model Dev 11 2209-2229.

Perugu H et al (2017) Atmos Environ 155 210-230.

Ramacher MOP, Karl M (2020) Int J Environ Res Public Health 17(6): 2099.

Singh V et al (2020) Environ Pollut 113623.

Strunk W, White EB (2011) The Elements of Style 4 Ed. Pearson.

Veratti G et al (2020) Atmos Environ 223 117285.

**Response to Reviewer 2**

The Authors are deeply honored to have the MS reviewed by reviewer 2. Please find our response on the corresponding comments below, with reference to the Reviewer's original comments in italics.

> **Comment:** *You should define the targets of your model somewhere at the beginning ... I hope you had the target group of readers in mind, when writing the paper.*

**Response:** The Author is correct. While Yeti can be used to generate aggregate data, and this was used as a basis for verification in §3.2, the main motivation behind its development is to generate emissions at road link resolution for use with urban and regional scale air quality models.

The title of the MS has been changed to "Yeti 1.0: a generalized framework for constructing bottom-up emission inventory from traffic sources at road link scale." Additional amendments have been made in §1 to emphasize the application on urban and regional scale modelling.

> **Comment:** *... the paper reads in many passages rather like a detailed user guide than a scientific paper.*

**Response:** This MS is submitted to GMD as a model description paper, which consists of the main paper, a set of user instructions and supporting outputs, with sufficient technical detail such that the methods can be reimplemented and for reproduction of corresponding results.

> **Comment:** *Together with the annexes and the supplementing information the text is extremely long for a description, how HBEFA emission factors have been multiplied with the number of vehicles passing street sections.*

**Response:** During the early and phases in Yeti development, it was recognized that data structure plays a much more critical role, where generality and performance are key. Thus the challenge in Yeti is to come up with a data representation that can handle input data of varying format and detail (i.e., especially traffic data), as well as a high computational throughput, as the amount of traffic data is typically large. These aspects have been implicitly described and discussed in §2.2 of the MS. The fact that Yeti can be perceived to be set of rudimentary arithmetic operations on the emissions factors and traffic activity is indicative of the effectiveness of its underlying data architecture, as pointed out in §9 of Raymond (1999).

> **Comment:** *… Why should the top down be "less sensitive to atmospheric transport processes and boundary conditions"? The calculated emissions do not depend on atmospheric transport. Please explain or adjust text.*

**Response:** The Reviewer is correct. The above statement was originally made in a different context and will be corrected in the revised MS.

> **Comment:** *How is the emission factor $\varepsilon$ selected from HBEFA? Traffic situation and level of service are needed as input for each road section. Did you perform a manual attribution of traffic situations for all roads in Berlin?*

**Response:** The emission factors are extracted from the public version (MS Access executable) of HBEFA, which are then selected in Yeti based on the traffic situation (TS) and the LOS of each vehicle subsegment. The attribution of TS and LOS has been performed as part of the input data for Yeti on the major road network of Berlin by the Berlin City Senate.

> **Comment:** *[I]s there any analysis behind setting the cold start share to 0.3? Please explain.*

**Response:** For the purpose of demonstrating compatibility between Yeti-generated aggregate traffic emissions and the official Berlin inventory, a cold start fraction of 0.3 was arbitrarily applied to all collector and access roads (HBEFA road types 40 to 50). The approach of applying cold start fraction is also seen in

Diegmann (2008) and it is anticipated that such information might not be available or collected in the traffic activity data. In the context of a model description paper, it is important to stress that Yeti can be configured (that is, without code change) to use different cold start fractions for all HBEFA road types in Yeti, should relevant information become available for a more systematic estimation of the cold start fraction.

Section 2.1.2 has been amended to indicate the similarity between treatment of cold excess emissions in Yeti and that in Diegmann (2008), and to emphasize user configurability for adaptation to other regions, particularly when cold start data are available.

> **Comment:** *Please describe how you included parking time. Did you use the 8 hour parking values only?*

**Response:** As one of the motivations for developing Yeti is to investigate effects of ambient conditions on traffic emissions, a design decision was made in Yeti to vary cold excess emission with the ambient temperature. As such, the other two variables – parking time and trip length – are assumed at a mean value. For the case study described in §§ 3.2 and 4, the mean values for Germany were selected, as per distribution indicated in the public version of HBEFA (under Info > Parking time / Trip length distributions). This is, of course, a limitation in Yeti resulting from using the public version of HBEFA, and ideally the determination of emission factors should be functions of all three variables. This limitation (of Yeti being reliant on the public version of HBEFA) is stated in §5.

> **Comment:** *Are deterioration factors for all exhaust components and ambient temperature corrections for hot NOx emissions from diesel cars considered? Same NOx emissions for winter and summer reported in chapter 4.1 suggest, that no temperature correction is included. Please specify possible simplifications.*

**Response:** The Reviewer is correct. For the demonstration of Yeti, only the base hot run emission factors are extracted from the public version of HBEFA 3.3 and 4.1 and used. As such, the corresponding modifications from mileage and temperature have not been considered explicitly. These adjustments affect free flow traffic (LOS 1) of gasoline- and diesel-fueled passenger vehicles traversing urban trunk roads (road type 21).

In the context of the present MS, the ability of Yeti to generate emissions that are dependent on ambient temperature has been shown based on the emission factors assimilated for its application. Should the HBEFA hot run emission factors be also extracted for use with Yeti, they would have been taken into consideration in the same manner as cold excess and evaporative emission factors. On a practical level, however, the fleet distribution data considered for the city of Berlin do not contain age or mileage information, as such the compensation of emission factors based on regular wear-and-tear cannot be quantified. The Authors agree that the inclusion of powertrain deterioration would be necessary to provide a more encompassing emission estimate, but it requires a more serious thought in design and implementation in order to anticipate and subsequently accommodate the possible ways in which mileage and age of the affected vehicle subsegments would be represented at a source data level.

Sections 3.1.1 and 3.1.3 have been amended to indicate that only base hot run emission factors are used for the purpose of demonstration, and that the Berlin traffic activity data, in particular the fleet distribution data, do not contain mileage-related information. In addition, the issue of ambient temperature dependence of hot run emissions will be introduced in §4.1. Both items will be identified as potential areas of future development in §5.

> **Comment:** *Equation 4: is the unit for the day to hour redistribution factor provided in B2 correct? Changing g/day into g/h suggests a unit of [day/hour] you state day$^{-1}$.*

**Response:** The Reviewer is correct. The units of the coefficients used in Equation (4), listed in Table 2, as well as Appendix B2, will be corrected.

**Comment:** *How you attribute the road category and hourly levels of service to all streets in Berlin on the other hand is more important but not described. You describe in chapter 3.1.3 input data, but do not mention any source for traffic flows, road categories etc.*

**Response:** The Authors also recognize the procurement of traffic activity being one of the most challenging aspects of preparing traffic emission inventories. Though the traffic activity provided to the Authors for generation of emission data with Yeti already contain road category, subsegment, and LOS information, etc.. In the meantime, the Authors took great care not to introduce additional constraints in incoming data, other than adherence to HBEFA basic conventions, such that traffic data can still be processed and provide insightful data in light of not incomplete information. As a result, the discussion outline in §3.1.3 is intended to serve as an illustrative example incoming traffic activity data of varying formatting and detail could be assimilated for use with Yeti, as opposed to be a predicate of required data.

Section 3.1.3 has been modified accordingly to emphasize this.

**References**

Diegmann V (2008) IMMIS/em/luft Version 4.0 User's Guide, IVU Umwelt GmbH.

Raymond ES (1999) The Cathedral and the Bazaar. O'Reilly, p 37.

**Response to Reviewer 3**

The Authors would like to thank the Reviewer for the meticulous comments. Please find our response on the corresponding comments below, with reference to the Reviewer's original comments in italics. In terms of minor comments, they will all be addressed unless otherwise stated.

> **General Comment 1:** *The level of detail that the framework needs for the input information and the resolution it can achieve are not clearly defined.*

**Response:** The ability to handle input data, in particular traffic activity, at a highly varying detail and input and format was part of the design of Yeti. Thus the input information and the achievable resolution are described to the extent that a prospective user can produce traffic emission data with Yeti that are representative of the available input information at hand. As a result, the discussion outline in §3.1.3 is intended to serve as an illustrative example incoming traffic activity data of varying formatting and detail could be assimilated for use with Yeti, as opposed to be a predicate of required data.

Section 3.1.3 has been modified accordingly to emphasize this.

> **General Comment 2:** *Currently the text reads very specific on file formats and model running processes. Suggestions to focus on the application of the framework and demonstrate the applicability are made.*

**Response:** This MS is submitted to GMD as a model description paper, which consists of the main paper, a set of user instructions and supporting outputs, with sufficient technical detail such that the methods can be reimplemented and for reproduction of corresponding results. A demonstration of the application of Yeti is provided for city of Berlin with results shown at an aggregate and road link level. This initial assessment can be found in §§3.2 and 4, respectively. We agree with the reviewer that an in-depth evaluation of these scenarios could be interesting but it is not the main goal of this publication. However, with the further use of Yeti there are current plans to pursue this line of work and present further analysis in subsequent publications.

> **General Comment 3:** *Some additional analyses are suggested to demonstrate the accuracy of the framework for all the modelled pollutants.*

**Response:** A verification of Yeti is made and §3.2 at an aggregate level, which showed comparable results with official inventory values. The accuracy of the Yeti generated traffic emissions can only be inferred by the aggregate results due to lack of direct measurements. Yeti employs HBEFA emission factors without modification, and thus its accuracy depends on the that of these emission factors, and the description of traffic activity. Both items are used as input, and their scrutiny is beyond the scope of a model description paper. However, as mentioned in the previous response, future studies are planned to integrate Yeti into existing air quality models and the accuracy of the emissions could potentially be evaluated against observational measurements.

Section 5 has been amended to reflect this.

> **Main Comment 1:** *Please specify what is meant by high spatial and temporal resolution in this study and justify how this aligns with existing literature (potentially with microscale emission modelling). For the spatial resolution, it would help to add the average length of the roads and the minimum and maximum lengths used in this study. For the temporal resolution, are the 1-hour intervals enough, have those been aggregated from more disaggregated information? Is that enough to capture traffic flow fluctuations that influence emission levels? Please specify and introduce in the text.*

**Response:** The introduction has been accordingly edited to address the above points. In particular: To the best of the Authors' knowledge and experience with urban-scale modelling (Chan and Butler, 2021, Khan

et al, 2021), the use of hourly traffic emissions at road link resolution (typically in the order of ~ 100 m), are considered state-of-the-art. Further, while some traffic emission modes, such as hot run emissions, can be obtained at higher temporal resolutions (i.e., PHEM), but the Authors deem this a risk to apply all modes of traffic emissions at such high resolution.

On the issue of road length, the Authors believe that not only the topology of the road network is relevant to the presentation and discussion of road side traffic emissions, as the Reviewer pointed out, but also the traffic flow through the road links in question. This information can be found in §4 and subsequently Appendix A for the relevant road sections (i.e., Frankfurter Allee and Silberstein Straße). On the other hand, the spatial distribution of Yeti-generated traffic emissions in Figure 5 have been normalized by the length of the respective road link so that the emission values are not confounded by the length of the individual road segments. Therefore, the presentation of statistics of road length (e.g., mean, maximum and minimum) to accompany of traffic emissions for the whole road network is no longer unnecessary.

> **Main Comment 2:** *A clear description of the added value of Yeti would help to understand better the utility, capabilities, interactions with existing knowledge and potential limitations of the proposed framework.*

**Response:** The HBEFA framework only provides emission factors. The onus is thus on the user to assemble all input data to which emission values could be produced at the highest achievable resolution for the input data. Yeti facilitates over a wide variety of input information in a systematic and consistent manner. While acknowledging existence of another commercial or in-house HBEFA-based tools that serve a similar purpose, the fact that Yeti is openly available to HBEFA users adds transparency into the process. Both arguments have been made in the introduction. On the other hand, Yeti carries limitations of using HBEFA emission factors. Some of these have already been addressed in §5 of the MS.

Additional limitations that the Reviewers pointed out, were also be included in §5.

> **Specific Comment 1:** *Page 2 line 29-30: Please specify which databases are meant when referring to "integration of vehicle-level emission factors estimated from existing databases.*

**Response:** This is intended to be a general, non-limiting introductory statement. Examples (i.e., HBEFA, COPERT, and MOVES) are given in the following paragraph.

> **Specific Comment 2:** *Page 2 line 49-51: Are COPERT, MOVES and HBEFA, considered in this study as high resolution emission models?*

**Response:** The use of COPERT and HBEFA for hourly, road-link resolution inventory or air quality models have been demonstrated in works such as Diegmann (2008), Ibarra-Espinosa et al (2018), Guevara et al, 2020, and Verrati et al (2020), which have all been referenced in the MS, particularly in the introduction. Meanwhile, Perugu et al (2017) discussed the possibility of incorporating their hourly, road-link resolution emission inventory model into MOVES, though such existing tool has not been identified, to the best of the Authors' knowledge.

> **Specific Comment 3:** *Page 2 line 54: Please be specific on the details and resolution the bottom-up approach can provide. In this case, specifying the resolution that can be achieved and the specific details that can be provided would help to understand better the capabilities and requirements of the bottom-up approach.*

**Response:** While affording higher temporal and spatial resolution (Guevara et al, 2017), the maximum possible temporal and spatial resolution afforded by the bottom-up approach are ultimately dependent on the resolution of the incoming data (Samaras et al, 1995; Coelho et al, 2014, Gurney et al, 2017).

The introduction of the MS was amended to include the above statement.

**Specific Comment 4:** *Page 3 line 65-72: Is Yeti a traffic emission inventory or a tool to calculate traffic emission inventories based on existing models and data? Please revise the text accordingly, in the current version it is not clear to me. Also, how flexible is the Yeti approach with the input information? Could it use second-by-second information and also with yearly averages? What are the limitations related to input information resolution. Since this framework seems to aim to work with varying levels of detail, did you analyze how the framework performed when using different levels of resolution for the input data?*

**Response:** The term "emissions inventory" (page 3 line 65 in the original MS) has been amended to "emission inventory framework". At the moment, the Authors can only create emission output on an hourly basis, but as a bottom-up inventory framework the emissions can be aggregated to longer time periods, as demonstrated in §3.2. For analysis and compatibility at varying levels of resolution, please refer to the Authors' response to Main Comment 1.

**Specific Comment 5:** *Page 3 line 71: Please explain the concept of "process symmetry", is not interpretable from the current text.*

**Response:** "Process symmetry" refers to identical and independent computational tasks performed over multiple processing units in a common operating system.

**Specific Comment 6:** *Page 3 line 77: What is meant by vehicle sectors and which specific ones have been investigated? Stating at this point a clear scope of the study would help to understand better the proposed framework capabilities and potential limitations.*

**Response:** "Vehicle sectors" (page 3 line 77) has been changed to "vehicle categories". While all vehicle categories have been considered in the study, exemplified in §§3 and 4, as well as Appendix A of the MS, the Authors find it premature to indicate this explicitly as an introductory remark as this is not relevant for the main goal of the paper which is the description of Yeti framework.

**Specific Comment 7:** *Page 3 line 76-80: The current text seems more as a description of what is done but makes not clear to me what are the goals to answer.*

**Response:** Please refer to Authors' response to General Comment 2.

**Specific Comment 8:** *Page 3 line 83: [H]ow long are the road segments in average? And what are the maximum and minimum length those could have for the proposed framework to estimate emission accurately?*

**Response:** Please refer to Authors' response to Main Comment 1.

**Specific Comment 9:** *Page 3 line 84: [W]here is the geometrical attribute of grade for each road segment coming from? Please specify in the text.*

**Response:** All general information on source data, including geometrical attribute of the road segment, can be located in Table 3 (page 25, line 650). The Authors find it impractical to display geometrical attributes on all road segments in the MS, however, as the major road network for the City of Berlin contains more than 10,000 road segments (and roughly 19,000 road links), as indicated in §3.2 onwards.

**Specific Comment 10:** *Page 3 line 84: [P]lease provide a definition for road segment, and road level. Also, along the text road link is mentioned. Please unify and/or provide [definitions].*

**Response:** The term "road level" (page 2, line 84) has been changed to "road link level". In addition, a brief explanation has been provided in the first paragraph of §2 to distinguish a "road segment" from a "road link."

**Specific Comment 11:** *Page 3 line 88: Please provide a definition for "vehicle subsegments".*

**Response:** In the context of the sentence it should be understood that "HBEFA vehicle subsegments" are being referred to, in conjunction to "HBEFA emission factors".

**Specific Comment 12:** *Page 5 line 120: Please provide a justification on why selecting only the temperature is the better option instead of selecting any of the other two options. [Did] you perform an analysis of the variables that have a stronger effect on cold start emissions or is this based on literature? If the first, please explain. If the second, please add the corresponding references. What is the associated error for the cold exhaust emissions calculation attributed to this selection?*

**Response:** HBEFA cold excess emission factors for a given vehicle subsegment are functions of ambient temperature, parking time, and trip length. The public version of HBEFA only allows those to vary with one of the three variables, while keeping the other two at predetermined average levels, where the average values for Germany are available. This has been indicated in §2.1.2.

As indicated in the introduction one of the motivations for developing Yeti is to investigate effects of ambient conditions on traffic emissions. Therefore, a design decision was made in Yeti to vary cold excess emission with the ambient temperature. Further, given the traffic activity information (i.e., traffic flow through each road link), information pertaining to parking time and trip length is not explicitly known, which further justified the use of surrogate mean values. This is, of course, a limitation in Yeti resulting from the public version of HBEFA, and ideally the determination of emission factors should be functions of all three variables.

This will be stated in §5.

**Specific Comment 13:** *Page 5 line 123-125: How are the cold start counts inferred from the hourly traffic count and the road type so that cold start events are identified? Please add explanation in the text.*

**Response:** The traffic activity data for Berlin, described in §3.1.3 do not contain information pertaining to the number of cold starts. It is also anticipated that this information might not be commonly available in many jurisdictions. An approach similar to Diegmann (2008) is implemented in Yeti, where a fraction of the traffic flow in a particular road link is assumed to be cold start traffic.

This reference will be added to §2.1.2.

**Specific Comment 14:** *Page 5 line 126-127: Why is the dimensionless factor set to 0.3? Please explain in the text the rationale behind selecting this specific number.*

**Response:** For the purpose of demonstrating compatibility between Yeti-generated aggregate traffic emissions and the official Berlin inventory, a cold start fraction of 0.3 was arbitrarily applied to all collector and access roads (HBEFA road types 40 to 50). In the context of a model description paper, it is important to stress that Yeti can be configured (that is, without code change) to use different cold start fractions for all HBEFA road types in Yeti, should relevant information become available for a more systematic estimation of the cold start fraction.

A comment will be introduced in §2.12 to emphasize this configurability.

**Specific Comment 15:** *Page 5 line 135-136: Please be specific with the terminology, I guess you are referring to emission factors from evaporative emissions[.]*

**Response:** The Authors are referring to "evaporative diurnal emissions" as stated in the text in question, which is defined in paragraph above (page 5, line 132).

**Specific Comment 16:** *Page 5 line 143-144: [W]hat is the share of fuel-injected vehicles Yeti assumes in the vehicle fleet? Is that a case-specific share? Please specify in the text.*

**Response:** Based on the context of the comment, the Authors assume that the Reviewer is referring to the fractions of fuel-injected and carbureted gasoline vehicles in the fleet. However, given that carburetors have been largely displaced by fuel-injection technologies since the 1990s, except in specialized applications, such as motorsports and small marine and household engines (both beyond the scope of the current paper), it is reasonable to accept that the contribution of evaporative diurnal emissions from carbureted vehicles is very small nowadays, when the overall contribution of evaporative emissions, based on the data presented in Table 7, is already about an order of magnitude smaller than the corresponding exhaust emissions.

Section 2.1.3 has been amended to state that evaporative emissions apply only to gasoline vehicles.

> **Specific Comment 17:** *Page 6 line 149: In this specific case, how many are the engine stops considered in HBEFA or how are those classified, and how does this translate to this specific study? Please, revise the manuscript for similar expressions.*

**Response:** The number of engine stops considered in Yeti depends on the function and the traffic activity on the road link in question, as indicated in Equation (6), in the same manner as cold starts in Equation (4) in §2.1.2.

> **Specific Comment 18:** *Page 6 line 151-152: What is referred to by direct data? Did you perform measurements? Please explain. Also, what percent of engine stops are coming, in this study, from direct data and what percent is estimated? Did you perform a comparison on how well the proposed estimation works compared to the "direct data" of number of engine stops?*

**Response:** Direct data refers to the number of measured engine stop events. This information is not available in the traffic activity data. Please refer to the Authors' response to Specific Comment 14 for additional details as the treatment of hot soak events is the same as that for cold starts. The Authors would like to further emphasize that, for the scope of the current MS as a model description paper, is that these currently incomplete (i.e., cold starts and hot soaks) can be accounted for in Yeti should direct data become available.

> **Specific Comment 19:** *For each of the emission processes (non-exhaust, evaporative, cold and hot exhaust), please specify which pollutants are estimated in Yeti.*

**Response:** This is implicit in Table 7, where contributions are broken down by each emission mode, where only applicable combinations (e.g., evaporative diurnal HC and non-exhaust PM) are presented. Further §2.1.4 already indicated to be applicable only to PM.

Section 2.1.3 will be amended to be applicable only to HC.

> **Specific Comment 20:** *Page 6 line 170: What are the implications of using two versions of the model to obtain different sets of emission factors for the different emission processes? Could the framework be adapted to be updated with current and coming versions of HBEFA to calculate emission inventories with the most updated emission factors available? The last is partially raised in the Summary at the end of the manuscript, but it seems that a specific update would be required with each update of HBEFA. An estimation on how fast Yeti can adapt to updates of the emission model would be useful.*

**Response:** The reason and implication of using HBEFA 4.1 on HBEFA 3.3 emission factors for non-exhaust PM is discussed in §3.2, where the results are summarized in Tables 6 and 7. The Authors speculate the deviation on non-exhaust PM emissions from officially reported levels under HBEFA 3.3 could be an issue with mapping HBEFA subsegments IDs between the two versions, which is described in §D1 of the supplement. As non-exhaust PM emission factors become available in the public version of HBEFA version 4.1 and onwards, users of Yeti for vehicle subsegments defined under HBEFA 4.1 and beyond will not face such issue. Additionally, Yeti is designed to work with existing HBEFA database infrastructure. So all is

needed is a viable version of HBEFA from which the corresponding emission factors are extracted; they can be used by Yeti without no code change.

**Specific Comment 21:** *Page 7 line 177: The term road links is used here. Different terminology is used throughout the text, previously road level, and road segments where mentioned. Please introduce a definition for each of them in the text.*

**Response:** Please refer to Authors' response to Specific Comment 10.

**Specific Comment 22:** *Section 2.2 seems to be very specific on the modules used, but I would be interested in knowing how those influence the emission calculation. For example, it is mentioned in lines 178-179 that Yeti can operate on a subset of the traffic network. Does that also imply that if for a specific subset very detailed traffic activity input information is available more detailed emission outcomes could be obtained? Can Yeti consider that in some way? How small can the network subset be? And on the other hand[,] what is the bigger network that would still be an option to run with Yeti. For the last, it would help if the authors provide a processing time and data size quantification for this specific study in Berlin.*

**Response:** The Python packages mentioned in §2.2 (`concurrent.futures` and `yaml`), as well as those listed in §A1 of the supplemental document, should already be part of the Python standard installation. They are mentioned here to emphasize the (a) that Yeti can take advantage of symmetric processes (see Authors' response to Specific Comment 5), while the "yaml" file format has been used for specifying Yeti configuration; and (b) the user will know what packages to install in the unlikely event that they are not part of the Python installation. Basic performance data is summarized on the paragraph starting line 307 in §3.2. The total storage required by Yeti for the city of Berlin is approximately 12.6 Gb, but this is not shown in the MS as it depends on the number road links, meteorological, diurnal traffic profiles, and fleet composition (i.e. the number of relevant subsegments).

On the issue of subset of traffic network, the Reviewer is correct. Using Yeti, the user can either operate on a subset of an existing set of traffic activity to save time, as exemplified in §4 of the MS, and later in Chan et al (2022a). The implication, as the Reviewer pointed out, highly detailed emission information can be achieved over a select number of road links where detailed traffic activity observation is available. The network subset must consist of, at a minimum, one road link.

**Specific Comment 23:** *Section 2.2.1 Data organization, section 2.2.2 User-specified configuration and section 2.2.3 Execution flow: th[ese] sections are lacking the specific relationship to the study done in Berlin. It kind of reads as a user manual of Yeti. For example[,] in line 195 it would help to know the specific execution parameters that were used in this study if relevant to explain some of the outcomes. Please revise, I think this is a good opportunity to show the applicability of Yeti and not place the focus only on how the framework internally works.*

**Response:** Please refer to Authors' response to General Comment 2.

**Specific Comment 24:** *Page 7 line 190: [S]everal new concepts are introduced in this line (emission strategy, day type, meteorological profile), please provide a definition for them.*

**Response:** "Emission strategy" is defined in §2.1 (entitled "Hourly emission calculation strategies"). The ability of Yeti to work with varying meteorological conditions has been discussed §§1 and 2, and the input data format for meteorological profiles is described in detail under §3.1.2 of the MS and §B5 in the supplement.

Amendments are made in §2.2.1 (page 7, line 191) to indicate what the possible day types can be.

**Specific Comment 25:** *Page 9 lines 241-243: [I]n reference to Table 2: some symbols from table 2 need additional explanation such as IDTS.*

**Response:** The caption for Table 2 will be amended to reflect that these are HBEFA table fields.

**Specific Comment 26:** *Page 9 Section 3.1.1. HBEFA emission factors and field data: please clearly state which ones are the field data specified in the title of this section and how those were obtained and used in Yeti. A note on how that could be replicated in other cities and which would be the needed information to obtain would be helpful.*

**Response:** A workflow on how the HBEFA data are to be obtained for Yeti are explained in §B3 and Table §S1 in the supplement. Including them in the main MS would make it much too long. These data will apply to regions where HBEFA emission factors are used. In addition, in the MS Berlin is used as a non-limiting illustrative example as seen in §3.1 (and particularly §3.1.3 for traffic activity) on how Yeti and corresponding HBEFA emission factors can be replicated in other regions. Due to the diversity in input data, in both detail and quality, the task of assimilating this information for particular jurisdiction must unfortunately lie in the prospective user.

**Specific Comment 27:** *Page 9 [S]ection 3.1.3. This section has a lot of details on how the files look like and the directories, but is lacking an explanation of the content of table 3. This also applies to previous section 3.1.1. Please, it would help to add an interpretation of the information presented in the tables to properly understand them. If not relevant for to explain the results in the main paper, maybe this part could be moved to the supplementary material.*

**Response:** This comment demonstrates the issue that arise from the multiplicity of input data format and details. As emphasized in previous responses, in developing Yeti, the Authors intend on giving the user a high degree of flexibility to adopt information from a wide variety of input data. In this context, the contents of Table 3 provides an example of what type of input data could be encountered, and how this information could be used in Yeti.

**Specific Comment 28:** *Page 10 line 268: Why would you have road segments with zero length? And what is the error associated to ignore the segments that have no indicated traffic direction? Are those a significant amount of your segments?*

**Response:** Road segments with zero length, ambiguous road directions, or no indicated traffic volume are erroneous entries in the network topology and thus not used in the determination of traffic data for Yeti.

This will be indicated in §3.1.3.

**Specific Comment 29:** *Page 10 line 274: please be specific with the highest resolution that can be achieved. How long are the individual road segments, and what would be the highest temporal resolution?*

**Response:** The sentence in question (i.e., page 10, line 274) indicates that "the highest resolution that can be achieved in the emission inventory is that of the individual road segment, within which vehicular distribution is assumed to be spatially uniform." Also, in the sentence prior to it stated the hourly nature of the vehicle count.

**Specific Comment 30:** *Page 10 line 280: [P]lease add the essential information to understand how the data recompilation was done. What period of time was covered with the measurements? What kind of roads were measured? How big was the sample size obtained and how were those postprocessed to be obtain a full dataset to use in Yeti?*

**Response:** The Authors do not fully understand the reason of this request. If the Reviewer refers to the recompilation of traffic activity data, it is beyond the scope of this MS, as the Authors only employed the traffic activity data as provided and rendered them into a format suitable for Yeti without modifying or creating new traffic activity data from it. As for assimilation of already existing traffic activity data, §3.1.2, as well as §B4 should contain sufficient information on how this is performed.

**Specific Comment 31:** *Page 10 line 280: Why was finally this attribution selected and how is this attribution of LOS IV vs LOS V affecting the results for this specific example?*

**Response:** The concept of LOS originates from HBEFA, as indicated in §3.1.1. Yeti simply uses the number of LOS defined in the particular version of HBEFA, that is, 4 LOS's in version 3.3 and 5 LOS's in version 4.1. The traffic activity data used in §§3.2 and 4 do not contain information on LOS 5, as the LOS distributions (see last entry in Table 3) is prepared prior to the release of HBEFA 4.1 and thus contains (still) only 4 LOSs. This also highlights the incompleteness of traffic activity data. Likewise, the official inventory is based on the same traffic data, which contains only 4 LOSs.

**Specific Comment 32:** *Page 12 line 356: [R]esults for the other pollutants (CO and PM) would be also interesting to see in the supplementary material. That would help to understand the applicability of the proposed framework, explain relationships between pollutants and accuracy of the outcomes for all the modelled pollutants in this study.*

**Response:** The inclusion of CO and PM in §3 is intended to be part of a comparison with the official inventory values. The decision to only focus NOx and HC in §§4.1 and 4.2 is mainly to highlight primary contributions and differences from season and road types. The Authors have decided to refrain from presenting additional data, at the risk of further changing the character of this MS from an already quite long description of Yeti and its underlying methodology to a scientific evaluation paper. The volume of comments from all reviewers on the scientific aspects of the input data presented in this MS already indicates such deviation in focus has taken place.

**Specific Comment 33:** *Page 13 section 4.1 (line 365): In figure 4, NOx emissions for summer and winter are the same. Can Yeti consider differences in traffic volumes and traffic composition for different seasons or even lower temporal resolutions (month, week, days,...)?*

**Response:** To be more specific, the NOx emissions on Frankurter Allee, a trunk road, remain the same in Summer and Winter. As indicated in §4.1 (page 13, line 364) indicated that this behavior, in the context of the HBEFA emission factors, is an expected behavior from Yeti due to the current configuration, where no cold start event is to take place on trunk roads. Further, as the Reviewer pointed out, this is also due to the fact that the traffic volume used are based on an annual average diurnal cycle segregated by day type. This issue was later observed in Chan et al (2022a), where deviations could be seen based on comparison of Yeti generated emissions and in-situ air quality measurements. Thus this follows, the consideration of traffic profiles of different traffic profiles, such as seasonal, monthly, weekly, or daily profiles, as indicated the Reviewer, was anticipated in the conception of Yeti and can be employed.

These two points, as the basis of this comment, will be incorporated into §4.1.

**Specific Comment 34:** *Page 13 Section 4.2 ... was fuel type also considered to analyze the outcomes? what is the average vehicle fuel distribution (% of diesel, gasoline,...) in the example roads for the different vehicle types (passenger cars, bus, motorcycle...)? Together with current tables A1 and A2, this additional table would help to understand better the outcomes presented in tables 10-11.*

**Response:** As HBEFA subsegment identifiers contain information on vehicle categories, emission classes, and power train technology, further contributions of Yeti generated emissions based on fuel type is possible. The Authors decided not to include this break-down because, while it is somewhat expected driven mainly by (no longer so) recent media interests surrounding diesel emissions manipulation, the results will be heavily dominated by gasoline and diesel, in terms of the fleet data in Berlin used in throughout the MS. Separating emission contributions based on vehicle categories and Euro emission classes, in the Authors' opinion, better showcase the ability of Yeti in reaggregating emission data in different ways, if such term is appropriate, and also yielded unexpected observations that serve as discussion points in §§4.1 and 4.2.

**Specific Comment 35:** *Page 15 Summary section: please revise to add notes on the scope and potential limitations of the framework raised previously.*

**Response:** Please refer to Authors' response to Main Comment 2.

**Specific Comment 36:** *Figure 5: add to the caption the average length of the segments.*

**Response:** Please refer to Authors' response to Main Comment 1.

**Minor Comment 1:** *Page 3, line 64: Revise English in the phrase "but the cost and the level configurability play a significant role".*

**Response:** The have Authors decided against changing this phrase at the risk of altering its nuance.

**Minor Comment 6:** *Page 13 line 383 and line 398:* **"***busses***".**

**Response:** "busses" is also an acceptable plural form for "bus".

**References**

Chan EC et al (2022a) Proc. HARMO 21 Aveiro (Portugal).

Coelho MC et al (2014) Sci Total Environ 470 127-137.

Diegmann V (2008) IMMIS/em/luft Version 4.0 User's Guide, IVU Umwelt GmbH.

Guevara M et al (2017) Air Qual Atmos Health 10 627-642.

Guevara M et al (2020) Geosci Model Dev 13(3), 873-903.

Gurney K et al (2017) Elem Sci Anth 5 44.

Ibarra-Espinosa S et al (2018) Geosci Model Dev 11 2209-2229.

Perugu H et al (2017) Atmos Environ 155 210-230.

Samaras Z et al (1995) Sci Total Environ 169 231-239.

Veratti G et al (2020) Atmos Environ 223 117285.